# Tackling immunosuppression by *Neisseria gonorrhoeae* to facilitate vaccine design

Rebekah A. Jones[1], Fidel Ramirez-Bencomo[2], Gail Whiting[3], Min Fang[3], Hayley Lavender[1], Kacper Kurzyp[1], Angela Thistlethwaite[2], Lenka Stejskal[2], Smruti Rashmi[2], Ann E. Jerse[4], Ana Cehovin[1], Jeremy P. Derrick[2], Christoph M. Tang[1] *

**1** Sir William Dunn School of Pathology, University of Oxford, South Parks Road, Oxford, United Kingdom, **2** School of Biological Sciences, Faculty of Biology, Medicine, and Health, University of Manchester, Manchester United Kingdom, **3** Medicines and Healthcare products Regulatory Agency, South Mimms, Potters Bar, Hertfordshire, United Kingdom, **4** Department of Microbiology and Immunology, Uniformed Services University, Bethesda, Maryland, United States of America

* christoph.tang@path.ox.ac.uk

**Data Availability Statement:** The authors confirm that all data underlying the findings are fully available without restriction. All relevant data are

## Abstract

Gonorrhoea, caused by *Neisseria gonorrhoeae*, is a common sexually transmitted infection. Increasing multi-drug resistance and the impact of asymptomatic infections on sexual and reproductive health underline the need for an effective gonococcal vaccine. Outer membrane vesicles (OMVs) from *Neisseria meningitidis* induce modest cross-protection against gonococcal infection. However, the presence of proteins in OMVs derived from *N. gonorrhoeae* that manipulate immune responses could hamper their success as a vaccine. Here we modified two key immunomodulatory proteins of the gonococcus; RmpM, which can elicit 'blocking antibodies', and PorB, an outer membrane porin which contributes to immunosuppression. As meningococcal PorB has adjuvant properties, we replaced gonococcal PorB with a meningococcal PorB. Immunisation with OMVs from *N. gonorrhoeae* lacking *rmpM* and expressing meningococcal *porB* elicited higher antibody titres against model antigens in mice compared to OMVs with native PorB. Further, a gonococcal protein microarray revealed stronger IgG antibody responses to a more diverse range of antigens in the *Nm* PorB OMV immunised group. Finally, meningococcal PorB OMVs resulted in a Th1-skewed response, exemplified by increased serum IgG2a antibody responses and increased IFNγ production by splenocytes from immunised mice. In summary, we demonstrate that the replacement of PorB in gonococcal OMVs enhances immune responses and offers a strategy for gonococcal vaccine development.

## Author summary

*Neisseria gonorrhoeae* is the bacterium that causes the sexually transmitted infection gonorrhoea. Gonorrhoea is a public health concern due to the bacterium developing resistance to numerous antibiotics, leading to the prospect of untreatable gonorrhoea infections. Untreated gonorrhoea causes severe complications, particularly impacting reproductive health. So far, no gonorrhoea-specific vaccine is available. *N. gonorrhoeae*

within the paper and its Supporting Information files.

**Funding:** This work was supported by Wellcome Trust Collaborative and Investigator Awards (214374/Z/18/Z and 221924/Z/20/Z to CMT). The funders had no role in study design, data collection and analysis, decision to publish, or preparation of the manuscript.

**Competing interests:** RAJ and CMT are co-inventors on a patent describing the use of gonococcal OMVs with meningococcal PorB.

vaccine efforts focus on optimising outer membrane vesicles (OMVs). However, the success of this approach may be hampered by the presence of immunomodulatory proteins in *N. gonorrhoeae* OMVs. Here we show that two key immunomodulatory proteins; the major outer membrane porin, PorB, and its stabilising protein, RmpM likely contribute to the lack of success of a gonorrhoea OMV-based vaccine. We improved immune responses to gonorrhoea OMVs by genetically modifying *N. gonorrhoeae* to produce OMVs lacking RmpM and containing a PorB replacement. Moving forwards, the modified OMVs represent a promising platform for a vaccine against gonorrhoea.

## Introduction

*Neisseria gonorrhoeae* causes the sexually transmitted infection gonorrhoea and has developed resistance against all known treatments [1]. When untreated, infection with *N. gonorrhoeae* can lead to severe complications, including pelvic inflammatory disease, infertility, ectopic pregnancy, and neonatal blindness [2]. Additionally, gonorrhoea facilitates the acquisition and spread of HIV [3]. Gonococcal infection is highly prevalent in low- and middle-income countries (LMICs), with the African WHO region estimated to have the highest annual incidence of gonorrhoea [4]. Therefore, the control of gonococcal infection would have a major impact on female sexual and reproductive health of those living in impoverished circumstances, exemplified by the disproportionate number of quality-adjusted life-years lost by women due to gonococcal infection [5]. *N. gonorrhoeae* has a remarkable ability to develop resistance to antibiotics through the acquisition of plasmids and chromosomal mutations [6]. Therefore, attention has turned towards preventative measures, including the development of vaccines [7]. Importantly, vaccines can play a key role in tackling AMR by decreasing the number of infections, thereby reducing antibiotic use and the emergence and spread of resistant bacteria [8].

There are several challenges faced in developing vaccines against *N. gonorrhoeae*. Importantly, natural infection does not appear to provide immunity, as *N. gonorrhoeae* suppresses adaptive immune responses and the killing mechanisms of innate immune cells [9]. *N. gonorrhoeae* is a human-specific pathogen, representing an additional challenge in developing physiologically relevant animal models of infection. Together these factors mean that there are no known correlates of protection against the gonococcus. Further challenges to vaccine development include the extraordinary capacity of *N. gonorrhoeae* to vary the composition of its cell surface through phase variation and antigenic variation [10], and the ability of the gonococcus to evade patrolling immune cells by residing intracellularly within epithelial cells, neutrophils, and macrophages [11,12]. Finally, *N. gonorrhoeae* utilises numerous mechanisms to manipulate the host environment to promote its survival, including suppressing host immune responses [13].

Immunosuppression induced by *N. gonorrhoeae* might be a significant factor when comparing the host response to infection with the gonococcus and other species of *Neisseria*. Asymptomatic colonisation of the nasopharynx by *Neisseria meningitidis* leads to the induction of an appropriate humoral response that prevents invasive meningococcal disease [14]. Equally, colonisation with the commensal species *Neisseria lactamica* can induce protection against infection with *N. meningitidis* [15]. However, infection with *N. gonorrhoeae* does not appear to elicit natural immunity against itself except in some highly exposed individuals [16]. In contrast, immunisation with *N. meningitidis* outer membrane vesicle (OMV)-based vaccines conferred a degree of cross-protection against gonorrhoea [17–19]; OMVs are naturally shed from the surface of Gram-negative bacteria and thereby contain a multitude of cell

surface proteins [20]. The effectiveness estimates of OMV-based meningococcal vaccines against *N. gonorrhoeae* range from 31% to 46%, which persists for up to three years post-vaccination [21]. Given the capacity of *Neisseria* species, other than *N. gonorrhoeae*, to elicit adaptive immune responses, it is perhaps unsurprising that meningococcal vaccines confer a degree of cross-protection against the gonococcus due to cross-reactive antigens.

A critical outstanding question is why *N. gonorrhoeae*-derived OMVs have been relatively unsuccessful as a vaccine platform. In Bexsero OMVs, only 25 outer membrane proteins common to *N. meningitidis* NZ98/254 and *N. gonorrhoeae* are present, of which only 12 are abundantly and consistently expressed in different batches [22]. Together with other fundamental differences in antigen expression, this suggests that *N. gonorrhoeae*-derived OMVs should have a greater efficacy against gonococcal infection than meningococcal OMVs. However, the presence of antigens within *N. gonorrhoeae* OMVs that suppress immune responses might undermine this approach. Importantly, *N. gonorrhoeae* suppresses the development of T helper (Th)1- and Th2-mediated adaptive immune responses, instead priming an innate immune response governed by Th17 cells, leading to neutrophil recruitment and promoting gonococcal survival [23]. Indeed, the most promising subunit and OMV vaccines for *N. gonorrhoeae* utilise adjuvants that drive a Th1 response, such as CpG and microencapsulated IL-12 [24,25]. Promoting a Th1 response is currently considered a key factor for a successful gonococcal vaccine, exemplified by intravaginal addition of IL-12 leading to faster clearance of murine gonococcal infection [26]. In mice, a Th1 response is characterised by interferon-γ (IFNγ) production, which stimulates the expression of the immunoglobulin (Ig) G2a antibody isotype, whereas a Th2 response is characterised by interleukin-4 (IL-4) production and stimulates the expression of IgG1 antibody isotype [27].

Two key antigens have proposed roles in *N. gonorrhoeae* immune evasion. The porin protein PorB is the most abundant outer membrane protein and facilitates ion exchange across the outer membrane. PorB is essential for the viability of *N. gonorrhoeae* and represents a large portion (~60%) of both outer membrane and OMV proteomes [28,29]. *N. gonorrhoeae* PorB has several immunomodulatory properties [30]. Gonococcal PorB contributes to resistance against the complement system by recruiting human negative complement regulators [31,32]. Gonococcal PorB also influences innate and adaptive immune responses by repressing cellular killing mechanisms and influencing apoptosis in neutrophils and macrophages, and by inhibiting T cell proliferation during antigen presentation [29,33]. Additionally, gonococcal PorB would not be an ideal vaccine antigen because it is extremely variable, with 1,229 unique PorB amino acid sequences reported across 5,000 gonococcal strains [34,35]. A second immune suppressive protein is RmpM; the N-terminus of RmpM binds to trimeric PorB in the bacterial outer membrane and links it to peptidoglycan through its C-terminal domain [36]. Antibodies against RmpM have been reported to block the recognition of other surface antigens, thereby preventing immune killing of bacteria [37].

*N. meningitidis* expresses two major porins, PorA and PorB, in contrast to other *Neisseria* species which only express a single major porin, related to PorB [38,39]. Importantly, meningococcal PorB displays broad immunostimulatory properties and has been used as a vaccine adjuvant due to its ability to induce antigen-specific B- and T-cell responses [40–42]. This is distinct from the properties of *N. gonorrhoeae* PorB that contribute significantly to immunosuppression [30]. PorB proteins from *N. meningitidis* and *N. gonorrhoeae* share 60–70% amino acid sequence homology, depending on the strains being compared [22,43].

Our objective was to generate a gonococcal OMV-based vaccine that lacks key immunomodulatory proteins which would otherwise suppress protective immune responses. As PorB is essential for the viability of *N. gonorrhoeae*, it has proven difficult to study the role of this protein during infection, including its influence on immune responses. To circumvent the

immunosuppressive properties of PorB and RmpM, we replaced *N. gonorrhoeae porB* with a meningococcal *porB* gene in a *N. gonorrhoeae* strain lacking *rmpM*; the introduction of meningococcal *porB* rescued the viability of gonococcus lacking its native *porB*. We show that mice immunised with OMVs derived from *N. gonorrhoeae* expressing meningococcal PorB elicit significantly higher antibody titres against a range of antigens compared to OMVs containing gonococcal PorB. We also show that immunisation with OMVs derived from *N. gonorrhoeae* expressing meningococcal PorB enhanced Th1-mediated responses compared to OMVs containing gonococcal PorB, exhibiting increased IgG2a antibody responses against OMV antigens and eliciting increased IFNγ from murine splenocytes. Our data suggest that *N. gonorrhoeae* PorB significantly impairs immune responses following administration as an OMV-based vaccine against this bacterium, and this can be circumvented by its replacement with meningococcal PorB, paving the way for the development of gonococcal vaccines derived from OMVs from the bacterium itself.

## Results

### Replacement of *porB*

We genetically engineered *N. gonorrhoeae* to remove the immunosuppressive properties of PorB and RmpM from our gonococcal vaccine. To generate *N. gonorrhoeae* FA1090 expressing a meningococcal *porB*, *N. meningitidis* MC58 *porB* ("*Nm* PorB") was amplified by PCR, fused to a kanamycin resistance cassette and introduced into FA1090 by transformation (Fig 1A). *N. meningitidis* MC58 *porB* is a class 3 *porB* allele while *N. gonorrhoeae* FA1090 *porB* is a PorB.IB allele [44]; the proteins share 68% amino acid identity (S1 Fig). Insertion of *Nm porB* was confirmed by PCR and sequencing, demonstrating that the gonococcal *porB* gene had been entirely replaced with the meningococcal gene. The resultant strain, FA1090$_{MC58\ PorB}$, had a reduced growth rate compared to wild type (WT) FA1090 (S2 Fig). Western blot analysis using PorB-specific typing monoclonal antibodies confirmed that FA1090$_{MC58\ PorB}$ expressed *Nm* PorB alone (Fig 1B). In addition, the *rmpM* gene was replaced with *ermC'*, and the deletion confirmed by PCR and Western blot analysis (Fig 1D). The deletion of *rmpM* exacerbated the growth defect of FA1090$_{MC58\ PorB}$ (S2 Fig).

### *Ng* PorB OMVs and *Nm* PorB OMVs have additional proteome differences

OMVs were generated from FA1090Δ*rmpM* (*Ng* PorB OMVs) and FA1090$_{Nm\ porB}$ Δ*rmpM* (*Nm* PorB OMVs). Upon analysis of the OMVs by denaturing SDS-PAGE, we observed different protein profiles of the *Ng* PorB OMVs and *Nm* PorB OMV preparations, in addition to the change in PorB. Differences in the OMV proteomes were consistent between independent batches (S3 Fig). To compare the OMV proteomes in detail, we analysed them by mass spectrometry. A total of 174 proteins were identified across three biological replicates of *Ng* PorB OMVs and *Nm* PorB OMVs (S2 Table). Proteins identified in only one sample, with low peptide number ($< 2$), or low overall coverage ($< 7\%$), were eliminated from the analysis, leaving 86 proteins identified with high confidence (FDR of $< 1\%$, S3 Table). Proteomic analysis confirmed that *Ng* PorB OMVs only contained FA1090 PorB, and *Nm* PorB OMVs only contained MC58 PorB. Other than PorB, one protein was found solely in *Nm* PorB OMVs, NGO_09965, an Opacity (Opa) family protein, and one protein was found solely in *Ng* PorB OMVs (NEIS0210, unknown function). A total of 31 out of the 86 proteins identified exhibited similar abundance in *Nm* PorB OMVs and *Ng* PorB OMVs, while 46 had a higher abundance (ratio $\geq 2$) in *Nm* PorB OMVs compared to *Ng* PorB OMVs, with only four proteins found to have lower abundance (ratio $\leq 0.5$) (Fig 2 and S3 Table).

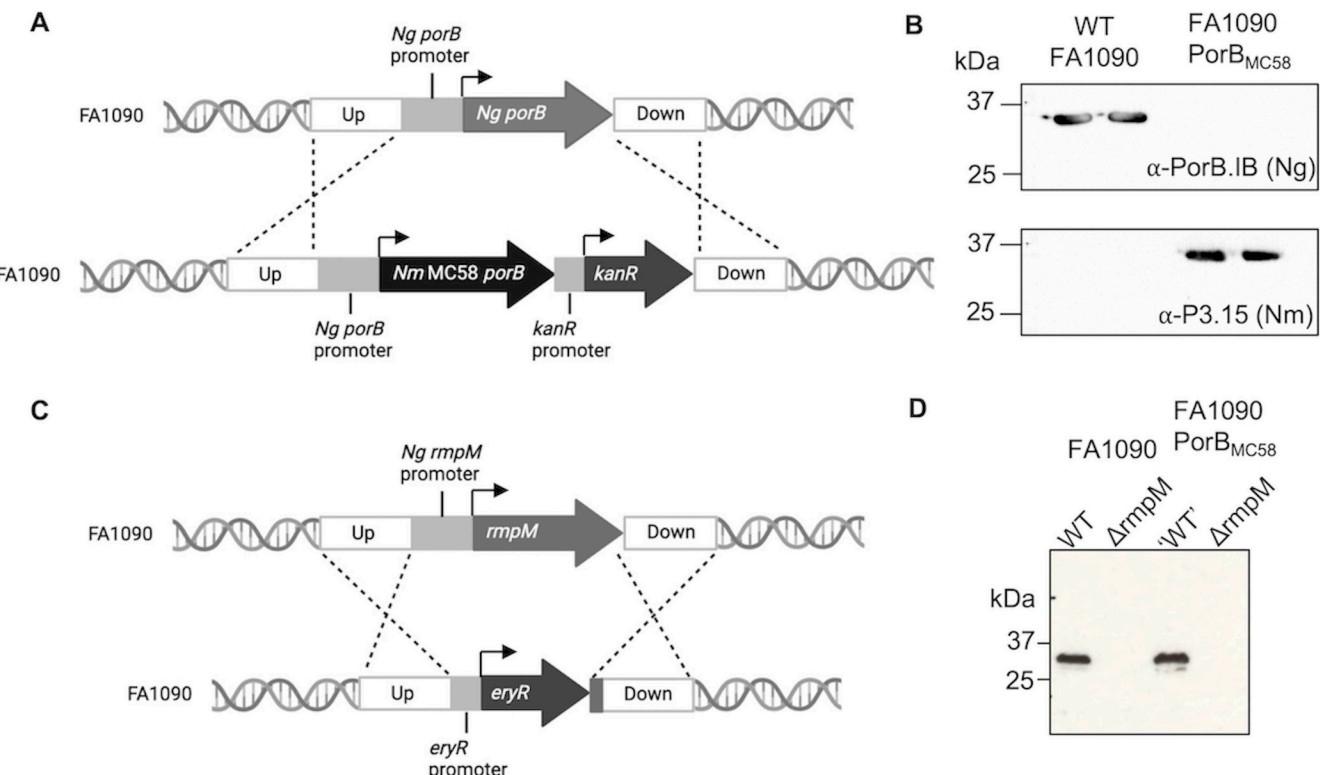

**Fig 1.** A) Generation of *N. gonorrhoeae* (*Ng*) FA1090 expressing MC58 *Nm porB*. Schematic representation of replacing the *Ng porB* gene with *Nm porB*, under the control of the native *Ng* promoter. Promoters shown in grey. Kanamycin resistance cassette (*kanR*) under its own promoter is present downstream of the open reading frame for selection. B) Western blots confirming the replacement of *Ng* PorB with *Nm* PorB, using mAbs against specific PorBs. C) Schematic representation of the deletion of *rmpM* in *Ng* FA1090 by replacement with an erythromycin resistance cassette (*eryR*). D) Western blot confirming deletion of *rmpM*. Arrows denote open reading frames. Dotted lines represent regions of homologous recombination. Figure not to scale.

Several groups of proteins with related functions exhibited a greater than two-fold increase in *Nm* PorB OMVs compared with *Ng* PorB OMVs. Iron acquisition and storage proteins transferrin binding protein B (TbpB), transferrin binding protein A (TbpA) and bacterioferritin (BrfB) were in higher abundance in *Nm* PorB OMVs. Furthermore, *Nm* PorB OMVs had a higher abundance of other nutrient acquisition proteins, including zinc-acquisition proteins TonB-dependent function protein-H and -J (TdfH, TdfJ), zinc-binding protein A (ZnuA, also known as MntC), and the methionine transporter MetQ. Potentially related to the increase in nutrient acquisition proteins, surface lipoprotein assembly modulator proteins 1 and 2 (Slam1 and Slam2) also had higher abundances; Slam1 is involved in the translocation of TbpB to the outer membrane [45]. The abundance of metabolism-related proteins ethanol–active dehydrogenase (AdhP), dihydrolipoamide acetyltransferase (AceF), and carbonic anhydrase (Cah) were also increased in *Nm* PorB OMVs. A final group of interest were proteins related to host-pathogen interactions, including *Neisseria* surface protein A (NspA), IgA1 protease (NEIS1959), macrophage infectivity potentiator (*Ng*-MIP), and *Neisseria* heparin binding antigen (NHBA), which were all increased in *Nm* PorB OMVs compared to *Ng* PorB OMVs.

Two Opa-related proteins, NGO_05420 and OpaD (NEIS0903) had the largest differences when comparing *Nm* PorB OMVs and *Ng* PorB OMVs, with abundance ratios (AR, *Nm*/*Ng*) of 74 and 44, respectively. Such a large difference in AR is consistent with ON:OFF phase variation, which is known to affect Opa expression [46]. An additional Opa54-related protein, NGO_06725, also had a higher abundance in *Nm* PorB OMVs, with an AR of 6.2.

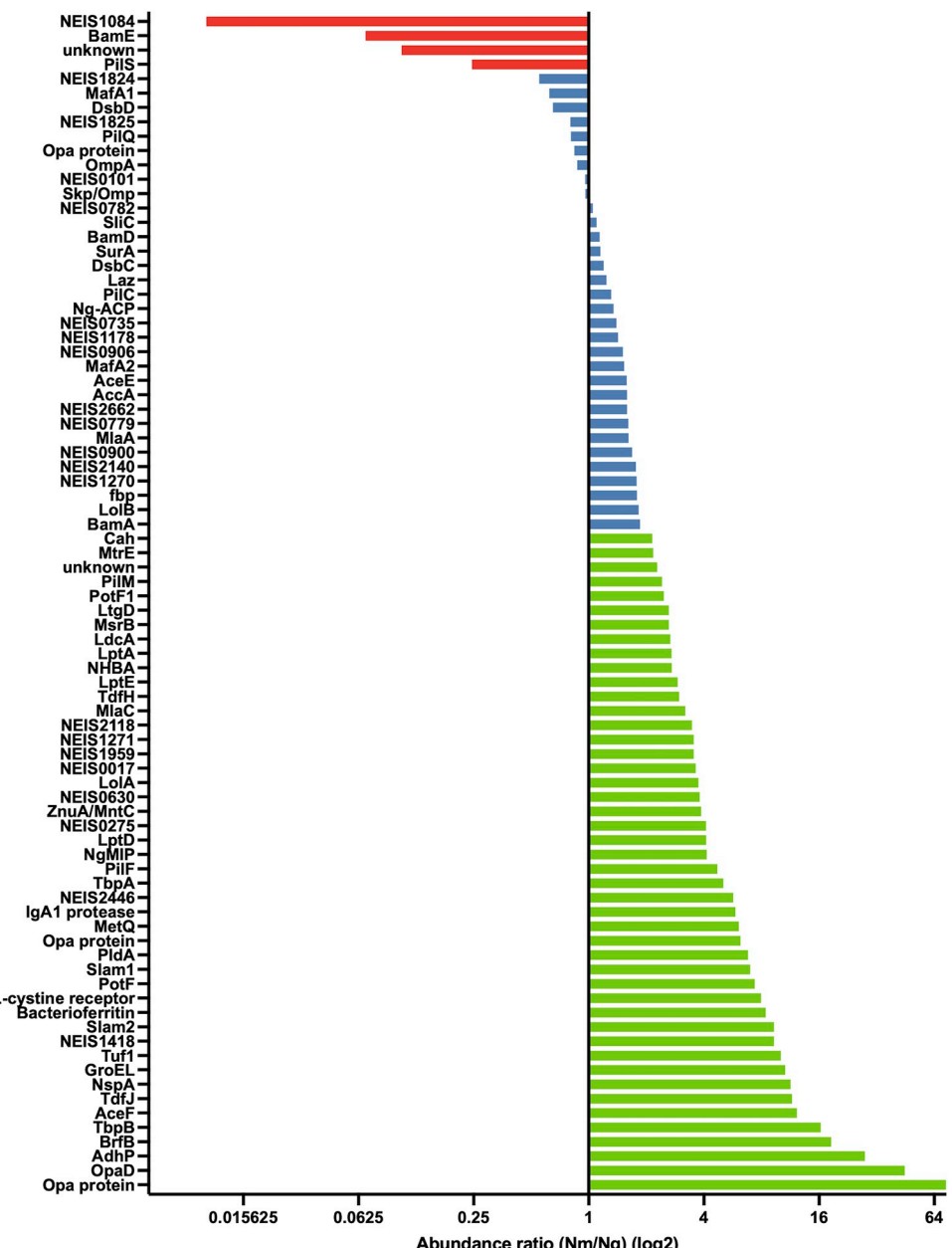

**Fig 2. The proteomes of OMVs derived from *N. gonorrhoeae* FA1090Δ*rmpM* and FA1090_MC58 PorBΔ*rmpM* differ.** Proteins with an abundance ratio (*Nm* PorB OMVs/*Ng* PorB OMVs) ≥ 2 are indicated in green, and those with an abundance ratio ≤ 0.5 are indicated in red. Proteins with an abundance ratio between 0.5 and 2 are indicated in blue.

Additionally, NGO_07725, an Opa54-related protein, was identified with slightly lower abundance in *Nm* PorB OMVs with an AR of 0.84. Together these data suggest that phase variation and/or mutations occurred within the Opa coding regions during the construction of the OMV-producing strains, resulting in the expression of multiple Opa54 proteins. Overall, replacing the *porB* gene in *N. gonorrhoeae* FA1090 with meningococcal *porB* resulted in additional differences in the OMV proteomes with changes in factors involved in nutrient acquisition, metabolism, and host-pathogen interactions.

**Table 1. Absolute quantification of PorB in *Ng*- and *Nm*-PorB OMVs using quantitative mass spectrometry.** Data are the mean ± standard deviation.

| Sample | PorB quantification (μg/mL) | Proportion of PorB in OMVs (%) | PorB in 12.5 μg immunisation (μg) |
|---|---|---|---|
| *Ng* PorB OMVs | 506 ± 191 | 70.2 ± 9.3 | 8.8 ± 1.2 |
| *Nm* PorB OMVs | 235 ± 67 | 36.5 ± 3.9 | 4.6 ± 0.5 |

As a significant number of proteins were more abundant in *Nm* PorB OMVs, we used quantitative mass spectrometry to determine whether the abundance of PorB differed between *Nm* and *Ng* PorB OMVs. PorB represented 70.2% (± 9.3%) of the *Ng* PorB OMV proteome, compared to 36.5% (± 3.9%) of the *Nm* PorB OMV proteome. Quantitatively, PorB measured 506 μg/mL (± 191 μg/mL) in *Ng* PorB OMVs and 235 μg/mL (± 67 μg/mL) in *Nm* PorB OMVs (Table 1). In a 12.5 μg vaccine dose, PorB would average 8.8 μg (± 1.2 μg) in *Ng* PorB OMVs and 4.6 μg (± 0.5 μg) in *Nm* PorB OMVs. In summary, the amount of PorB is reduced in *Nm* PorB OMVs compared to *Ng* PorB OMVs, potentially reflecting the increased abundance of other proteins.

## OMVs from FA1090$_{Nm\ porB}\Delta rmpM$ generate a higher antibody titre in mice

To determine the effect of replacing gonococcal PorB with meningococcal PorB on immunogenicity, mice were immunised with *Ng* PorB OMVs or *Nm* PorB OMVs together with a heterologous antigen, fHbp. *N. meningitidis* fHbp does not share homology to any protein in *N. gonorrhoeae* OMVs and was administered at a fixed amount. Anti-fHbp IgG antibody titres were significantly higher when mice were immunised with fHbp with *Nm* PorB OMVs when compared with *Ng* PorB OMVs, with endpoint titres ($log_{10}$) of 6.2 and 4.7, respectively ($p < 0.001$, Fig 3A), demonstrating that immune responses against an exogenous protein were higher in the presence of *Nm* rather than *Ng* PorB OMVs. Antibody responses against MtrE and MetQ, candidate antigens in gonococcal vaccines [24, 47], were also examined. MtrE is a surface-exposed component of a drug efflux pump which is upregulated in AMR strains of *N. gonorrhoeae*, while MetQ is involved in methionine uptake [24, 47]. Anti-MtrE and anti-MetQ IgG antibody titres were also significantly higher in mice immunised with *Nm* PorB OMVs compared to *Ng* PorB OMVs, with endpoint titres ($log_{10}$) of 5.6 and 4.7 for MtrE, and 5.8 and 4.2 for MetQ, after immunisation with *Nm* PorB OMVs or *Ng* PorB OMVs, respectively ($p < 0.001$, Fig 3B and 3C).

To further understand the difference in IgG responses, the ratios of IgG1 to IgG2a antibody titres were examined, as an indirect measure of Th1 *vs.* Th2 responses [27]. IgG subclasses differed in their titres when comparing antibody responses to recombinant fHbp, as well as the two OMV-based antigens, MtrE and MetQ. For fHbp, IgG1 antibodies comprised a higher proportion of the total IgG titre after immunisation with *Nm* PorB OMVs, with an endpoint titre ($log_{10}$) of 4.5 (Fig 3D). In contrast, for MtrE and MetQ, IgG2a antibodies represented a higher proportion of total IgG responses after immunisation with *Nm* PorB OMVs, with endpoint titres ($log_{10}$) of 3.7 and 4.9, respectively (Fig 3E and 3F). When examining the IgG1/IgG2a ratios to indicate a Th1- or Th2-skew, immunisation with *Ng* PorB OMVs resulted in a Th1-bias for all three antigens examined, with IgG1/IgG2a ratios of 0.4, 0.3 and 0.6 for fHbp, MtrE and MetQ, respectively (Table 2). Immunisation with *Nm* PorB OMVs resulted in a Th2-bias against fHbp, with an IgG1/IgG2a ratio of 4.1, but a stronger Th1-bias against the OMV antigens MtrE and MetQ, with ratios of 0.1 and 0.05, respectively. Overall, *Ng* PorB OMVs elicited responses against fHbp and OMV antigens with a consistent, although small, Th1-bias. However, *Nm* PorB OMVs elicited a marked Th1-bias against OMV antigens, but a Th2-dominated response against the recombinant antigen, fHbp.

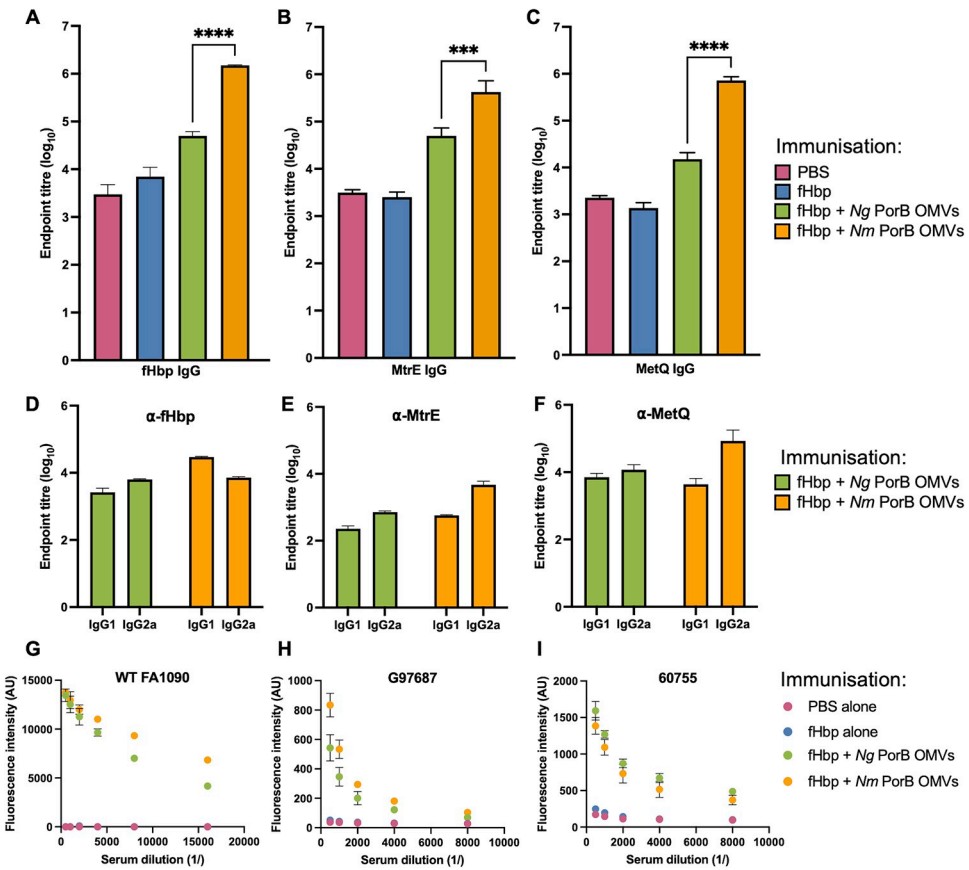

**Fig 3. Murine antibody responses after immunisation with *Ng*- or *Nm*-PorB expressing OMVs.** A-C) Endpoint IgG antibody titres of pooled sera recognising the model antigen fHbp, and the OMV antigens MtrE and MetQ, determined by ELISA. D-F) Endpoint IgG1 and IgG2a titres recognising the model antigen fHbp, and the OMV antigens MtrE and MetQ, determined by ELISA. Data are the mean ± standard deviation. *** p < 0.001, **** p < 0.0001. G-I) Antibody deposition on whole cell *N. gonorrhoeae* strains FA1090, G97687 and 60755 determined by flow cytometry showing the mean fluorescence intensity.

As higher antibody titres were observed against individual protein antigens MtrE and MetQ after immunisation with *Nm* PorB OMVs compared to *Ng* PorB OMVs, flow cytometry was used to investigate serum IgG antibody binding to whole cell *N. gonorrhoeae*. Serum from mice immunised with *Nm* PorB OMVs exhibited higher IgG antibody deposition against wild-type *N. gonorrhoeae* FA1090 when compared to *Ng* PorB OMV immunised mice (Fig 3G), even though FA1090 expresses the same PorB as in the *Ng* PorB OMVs that were used for immunisation. Antibody binding to two clinical isolates was also investigated, a multi-drug resistant strain G97687 (which expresses PorB.IB) and an isolate from Kenya, 60755 (which

**Table 2. The ratio of IgG1 to IgG2a geometric mean antibody titres to fHbp, MtrE and MetQ, elicited after immunisation with *Nm*- or *Ng*-PorB OMVs.**

| Antigen | Immunisation: *Ng* PorB OMVs | Immunisation: *Nm* PorB OMVs |
|---|---|---|
| fHbp | 0.412 | 4.068 |
| MtrE | 0.318 | 0.121 |
| MetQ | 0.597 | 0.052 |

expresses PorB.IA) [48, 49]. For strain G97687, serum from *Nm* PorB OMV immunised mice had higher IgG antibody deposition compared to serum from *Ng* PorB OMV immunised mice (Fig 3H). For strain 60755, IgG antibody binding was similar between the two OMV-immunised groups (Fig 3I).

As naturally released OMVs containing unmodified lipooligosaccharide (LOS) were used in immunisations, murine immune responses to LOS were investigated. Western blot analysis of LOS extracted from *N. gonorrhoeae* WT FA1090 or the LOS mutant FA1090Δ*lgtE* with sera from mice immunised with *Ng* or *Nm* PorB OMVs showed anti-LOS antibody responses in both immunisation groups (Fig 4A). A smaller LOS species was observed in the LOS mutant FA1090Δ*lgtE* compared with the wild-type strain, consistent with LOS truncation in the *lgtE* mutant and suggesting that OMVs can elicit antibodies specific to regions deeper within the LOS structure. Further analyses focused on responses against the LOS 2C7 epitope, which is the target epitope of candidate gonococcal vaccines [50]. Western blot using an anti-2C7 monoclonal antibody (mAb) [51] confirmed the 2C7 epitope was present in both *Ng* and *Nm* PorB OMVs, as well as WT FA1090 as demonstrated previously [52] (Fig 4B). As expected, 2C7 binding was not detected in the FA1090Δ*lgtE* LOS mutant. Next, an ELISA was used to determine whether *Ng* and *Nm* PorB OMV immunised mice had generated antibodies against

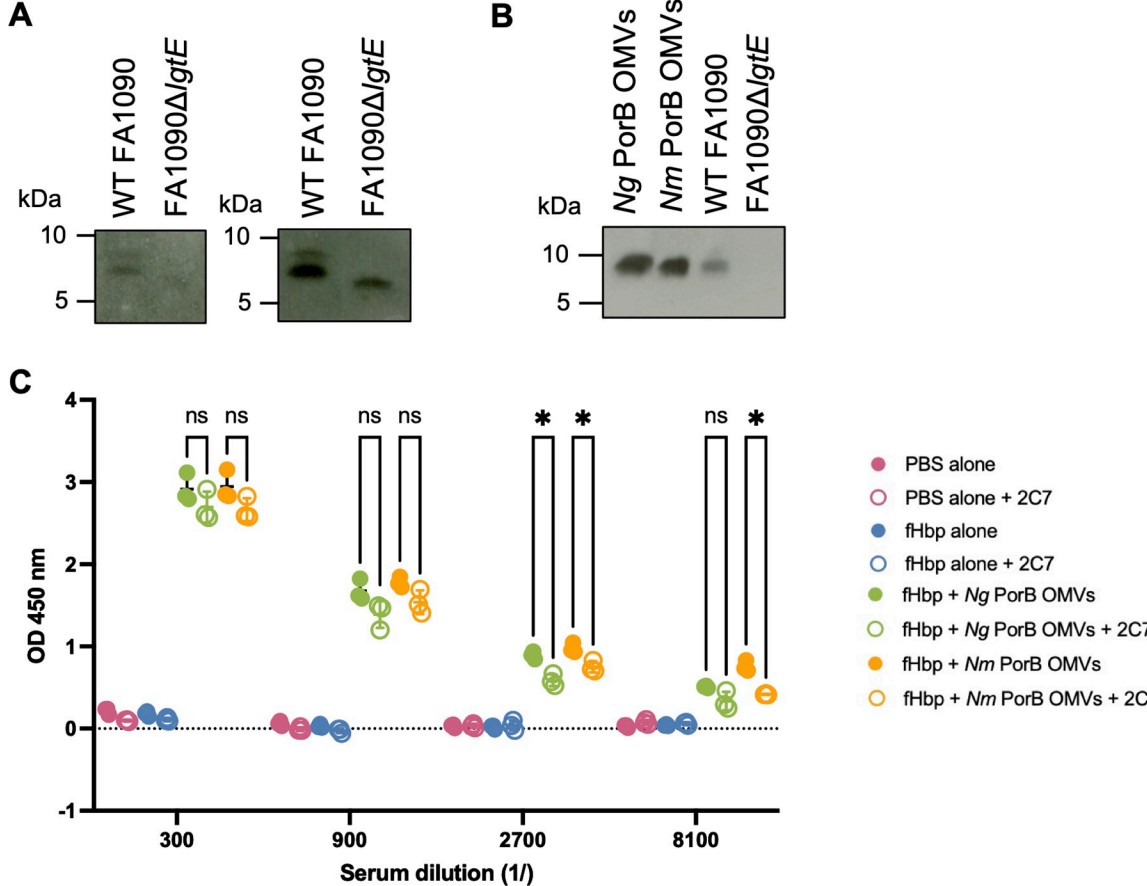

**Fig 4. Murine antibody responses to gonococcal lipooligosaccharide (LOS) after immunisation with *Ng*- or *Nm*-PorB expressing OMVs.** A-B) Western blots probing gonococcal LOS with murine sera after immunisation with *Ng* PorB OMVs (left blot) or *Nm* PorB OMVs (right blot) (A) or with 2C7 mAb (B). C) LOS ELISA probing with murine sera from *Ng*- or *Nm-PorB* OMV immunised mice or control groups, with or without pre-blocking with the 2C7 mAb. * *p* < 0.05.

**Table 3. Murine serum bactericidal activity[a] against *N. gonorrhoeae* strains FA1090, G97687 and 60755 after immunisation with *Ng*- or *Nm*-PorB OMVs.**

| Immunisation group | SBA titre against FA1090[b] | SBA titre against G97687[b] | SBA titre against 60755[b] |
|---|---|---|---|
| PBS alone | No killing | No killing | No killing |
| fHbp alone | No killing | No killing | No killing |
| *Ng* PorBΔ*rmpM* OMVs + fHbp | 4000 (4000, 4000, 2000) | 500 (500, 500, 500) | 1000 (1000, 1000, 500) |
| *Nm* PorBΔ*rmpM* OMVs + fHbp | 2000 (2000, 2000, 2000) | 500 (500, 1000, 500) | 500 (500, 500, 500) |

[a]Serum pooled from five mice was tested for bactericidal activity against *N. gonorrhoeae*, and the data are presented as the reciprocal of the highest dilutions where ≥50% of killing was observed. The median values from three biological replicates are indicated.

[b]IgG- and IgM-depleted normal human sera was used as the complement source and was required at a final concentration of 3% for FA1090 and 10% for G97697 and 60755.

the 2C7 epitope. ELISA plates coated with LOS from WT FA1090 were first blocked with the 2C7 mAb before probing with sera from mice immunised with *Ng* or *Nm* PorB OMVs. As both the 2C7 mAb and the OMV-immunised sera are murine, we utilised the fact that the 2C7 mAb is an IgG3 isotype and used anti-IgG1 and -IgG2a secondary antibodies to detect binding of antibodies in OMV-immunised sera. A small yet significant decrease in LOS binding was observed for both *Ng* and *Nm* PorB OMV immunised groups when the 2C7 epitope was blocked by adding the mAb (Fig 4C), indicating that immunisation with naturally released OMVs elicits 2C7-specific IgG1 and/or IgG2a antibodies.

To assess the functionality of the antibody responses, serum bactericidal assays (SBA) were performed using pooled sera from five OMV-immunised mice. *N. gonorrhoeae* FA1090 was incubated with serial dilutions of heat-inactivated murine sera before the addition of IgG- and IgM-depleted normal human sera (NHS) as the complement source. Human complement-dependent SBA titres (inverse of the serum dilution giving ≥50% killing of *N. gonorrhoeae*) were similar for *Ng* PorB OMV sera and *Nm* PorB OMV sera against FA1090, only differing by a single dilution (Table 3 and S4A Fig). SBA titres were also determined using *N. gonorrhoeae* clinical isolates G97687 and 60755. Both *Ng* PorB OMV sera and *Nm* PorB OMV sera exhibited SBA against G97687 and 60755, with similar titres (Table 3 and S4B–S4C Fig). Overall, immunisation with *Ng* or *Nm* PorB OMVs elicited bactericidal antibodies in BALB/c mice that were cross-reactive against clinical isolates of *N. gonorrhoeae* that have different antigenic profiles.

## Immunoprofiling of murine antibody responses against gonococcal antigens

To further characterise antibody responses after immunisation with OMVs, sera from individual immunised mice were used to probe microarrays containing 91 gonococcal surface proteins. The serum IgG reactivity to each individual gonococcal antigen was determined for mice immunised with i) PBS alone, ii) fHbp alone, iii) fHbp with *Ng* PorB OMVs, or iv) fHbp with *Nm* PorB OMVs and shown as a heat map or volcano plots (Figs 5 and S5, respectively). Low background responses for the PBS and fHbp-immunised groups are apparent. For sera derived from mice immunised with OMVs, reactivities against multiple gonococcal antigens were detected. When comparing the *Ng* PorB OMV and *Nm* PorB OMV immunised groups, several antigens exhibit reactivities in both groups. Antigens with stronger reactivities indicated by higher mean fluorescence intensity (MFI) included MtrE, SliC, Lipoprotein 2 (NEIS0906), GNA2091 (NEIS2071), NEIS1462, Lipoprotein 1 (NEIS1063), PilQ and

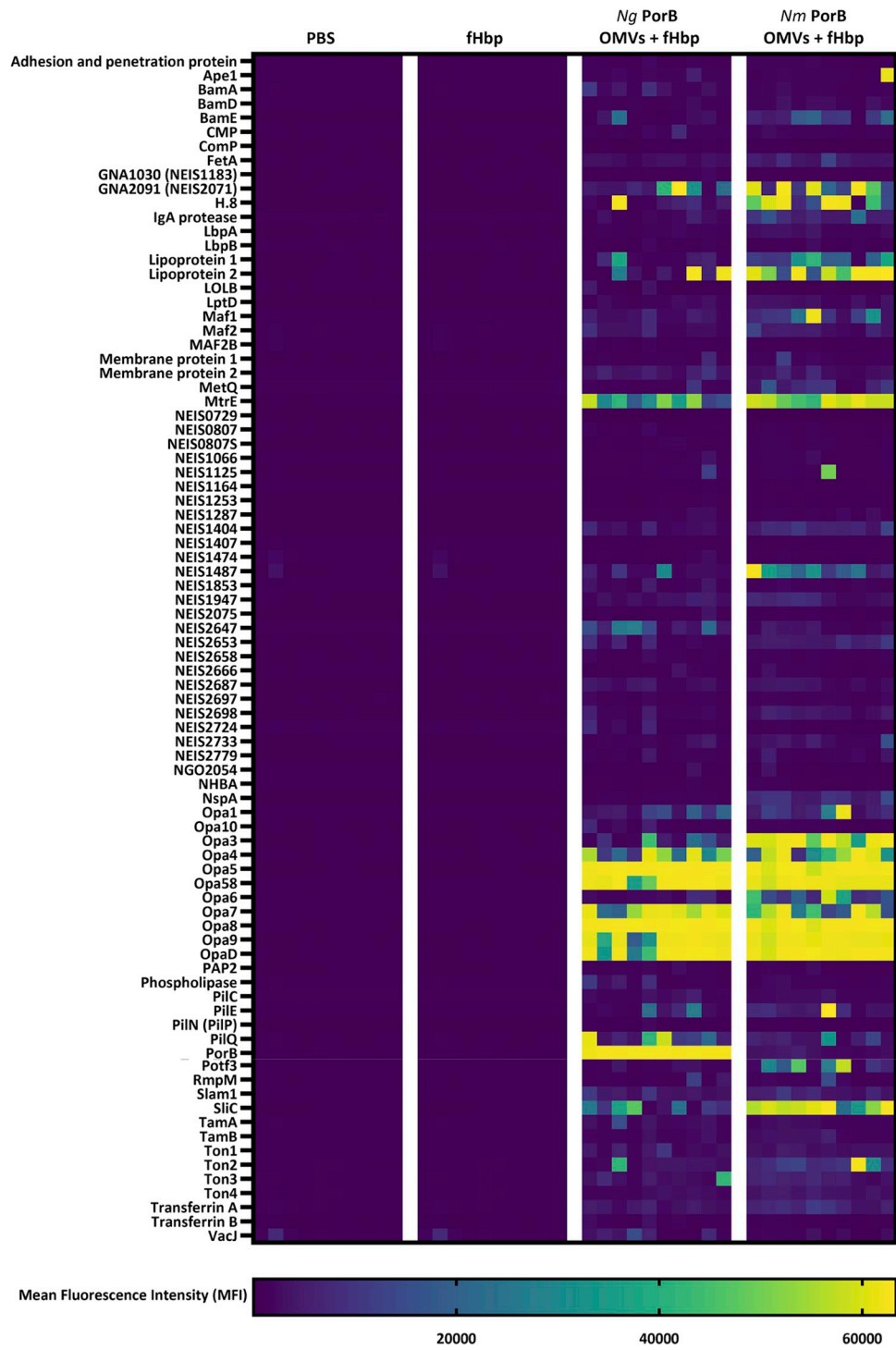

**Fig 5. Antigen-specific total IgG responses in sera from individual mice immunised with *Ng* PorB OMVs (third section) or *Nm* PorB OMVs (fourth section), compared to PBS alone (first section) and fHbp alone (second section); each column is the result for an individual mouse.** Individual antigens are in indicated in each row. Colour represents the mean fluorescence intensity (MFI) value for antibody binding to each antigen.

NEIS1487. Of these eight proteins, the antibody reactivity was stronger in the *Nm* PorB OMV immunised group for seven of the antigens. PilQ was the exception, where reactivity in *Ng* PorB OMV immunised mice was stronger. A relatively lower MFI was observed in both OMV immunised groups for BamE, PilE, and Ton2, where again the reactivity was higher in the *Nm* PorB OMV immunised group for three of the four antigens; for Slam1, reactivity levels were similar between *Nm* PorB OMV and *Ng* PorB OMV immunised mice. A subset of antigens showed reactivity in only one group; NEIS2647 (NGO554) displayed reactivity in only *Ng* PorB OMV immunised mice, whereas a larger number of antigens, including Ape1, IgA protease, Maf1, MetQ, NspA, Potf3, NEIS1125 and NEIS1405, displayed reactivity only in *Nm* PorB OMV immunised mice. The stronger antibody responses to MtrE and MetQ in *Nm* PorB OMV immunised mice, compared to *Ng* PorB OMV immunised mice, shown by the protein microarray are consistent with the ELISA data (Fig 3). Overall, microarray data analysis demonstrated that total murine IgG antibody responses were higher and more diverse after immunisation with *Nm* PorB OMVs compared to immunisation with *Ng* PorB OMVs.

The gonococcal protein microarray revealed strong reactivities to PorB and Opa variants. The microarray included PorB from FA1090 and, as expected, mice immunised with *Ng* PorB OMVs generated a strong antibody response to FA1090 PorB, the variant present in these OMVs. In contrast, *Nm* PorB OMVs elicited very weak antibody responses to FA1090 PorB. For Opa proteins, both *Nm* PorB OMV and *Ng* PorB OMV immunised mice exhibited widespread reactivity against Opa variants. The reactivity profiles to Opa variants are remarkably similar for both OMV immunised groups, particularly given the differences in Opa expression profiles revealed by proteomic analysis of the OMVs, consistent with responses elicited against epitopes shared between Opas.

To further assess murine polyclonal IgG responses after immunisation with *Ng* or *Nm* PorB OMVs, the gonococcal protein microarrays were also used to analyse the reactivity of IgG1 and IgG2a subclasses (Fig 6). Similar prominent profiles for IgG1 and IgG2a reactivity to the Opa variants were observed for both *Ng-* and *Nm-*PorB OMVs. Interestingly, some antigens tended to elicit IgG2a over IgG1 antibody responses; for example, BamE, SliC, Lipoprotein 1 (NEIS1063) and Lipoprotein 2 (NEIS0906) have relatively stronger MFI signals for IgG2a. However, this observation was independent of the *Ng/Nm* PorB OMV immunisation group. Overall, when examining individual antigens, reactivity in *Nm* PorB OMV immunised mice was stronger for both IgG1 and IgG2a, compared to *Ng* PorB OMV immunised mice, suggesting that increases in both subclasses contributed to the overall higher IgG responses shown in Fig 5.

Principal Component Analysis (PCA) was applied to all four immunisation groups in order to capture the variance between each serum sample and group samples with similar reactivity profiles. As both OMV groups responded strongly to nearly all Opa proteins and only *Ng* PorB OMVs elicited PorB reactivity, we removed PorB and Opa variants to ascertain which other antigens were contributing to the different responses after immunisation with *Ng* or *Nm* PorB OMVs. PCA of individual serum samples showed that *Ng* PorB OMV and *Nm* PorB OMV immunised mice were well separated from the fHbp alone and PBS control groups in the PC1 dimension (Fig 7A; each point is a serum sample from a single mouse). Even with the PorB and Opa variants removed, sera from *Nm* PorB OMV immunised mice were separated from *Ng* PorB OMV serum samples, further from the PBS controls. This observation is attributable to a greater amplitude of antigen responses overall in the *Nm* PorB OMV serum samples. The antigens contributing most strongly to this separation were Lipoprotein 2 (NEIS0906), outer membrane protein H.8, Potf3, Lipoprotein 1 (NEIS1063), SliC, MtrE, GNA2091 (NEIS2071) and NEIS1487 (Fig 7B). In summary, PCA showed that differences in murine IgG antibody

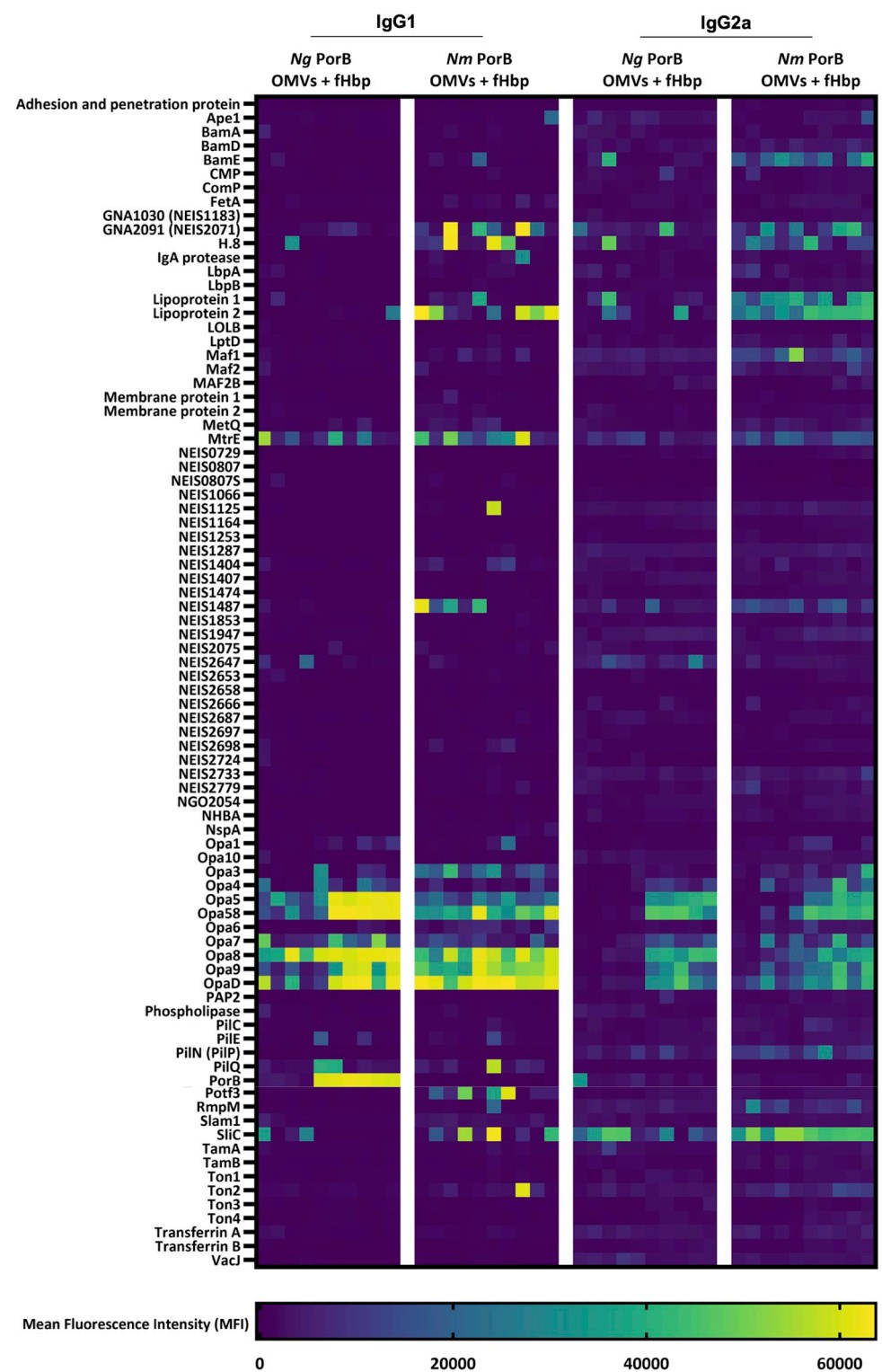

**Fig 6. Antigen-specific IgG1 and IgG2a responses in sera from mice immunised with *Ng* PorB OMVs or *Nm* PorB OMVs.** Individual antigens are in indicated in each row; each column is the result for an individual mouse. Each colour block represents the mean fluorescence intensity (MFI) value for antibody binding to each antigen.

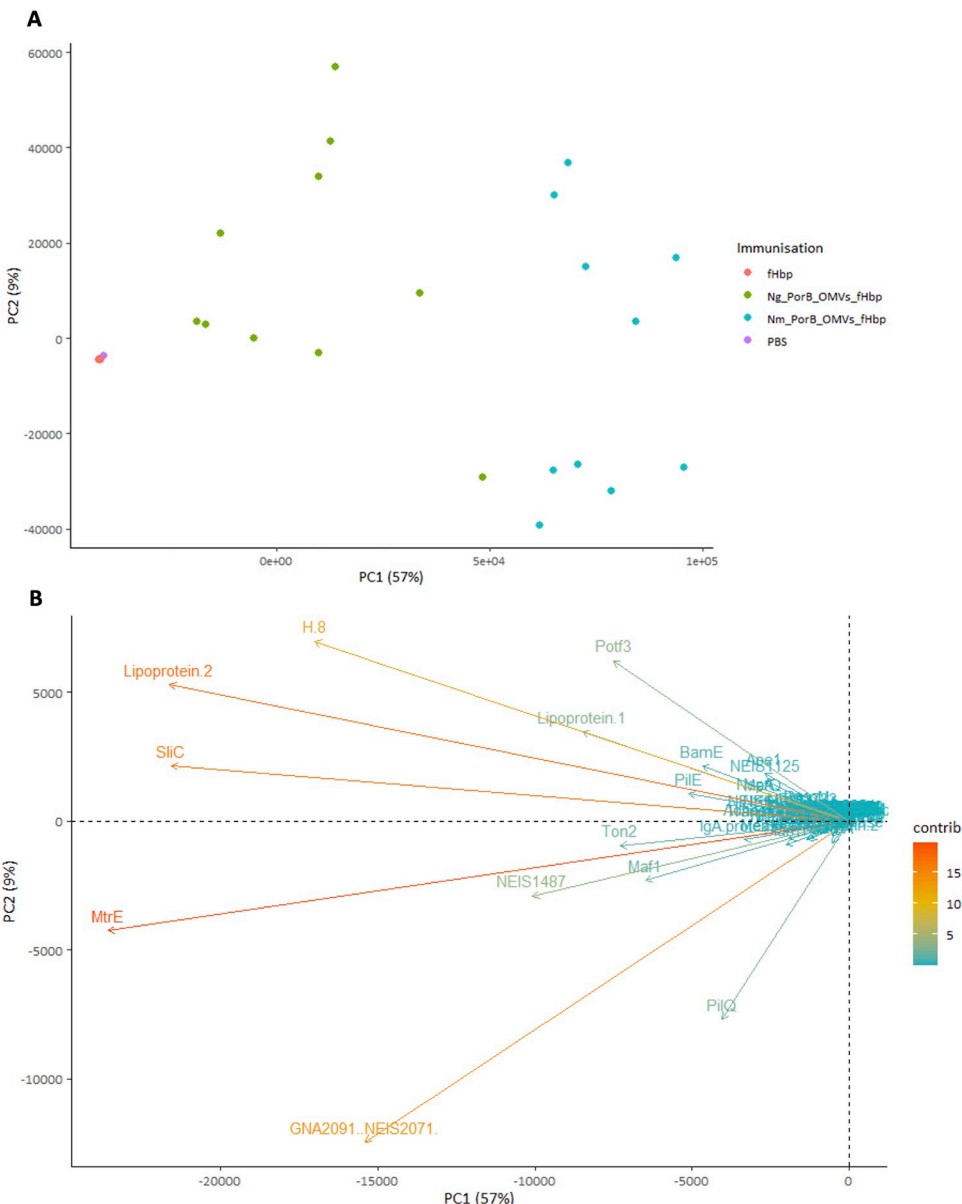

**Fig 7. PCA applied to antigen-specific total IgG responses.** A) Separation by individual serum sample, including all antigens. The biplot is generated using the squared coordinates (cos²) for PC1 and PC2, calculated as the squared coordinates of the eigenvalues. B) Contributions of individual antigens, excluding PorB and Opa variants, to PC1 and PC2 separation.

responses to *Ng* PorB or *Nm* PorB OMVs are attributable to several different gonococcal antigens.

## *Nm* PorB OMVs elicit pro-inflammatory responses in *ex vivo* stimulated splenocytes

As the switch from *Ng* PorB to *Nm* PorB in gonococcal OMVs enhanced humoral immune responses, we investigated cellular immune responses elicited by *Ng*/*Nm* PorB OMVs by re-stimulating splenocytes from immunised mice with the vaccine antigens. Production of the

pro-inflammatory cytokine IFNγ, the effector cytokine indicative of a Th1 response [53], was significantly higher in splenocytes from *Nm* PorB immunised mice when compared to the *Ng* PorB OMV immunised group, when re-stimulated with either *Ng* PorB OMVs ($p < 0.05$) or *Nm* PorB OMVs ($p < 0.001$) (Fig 8A). Overall, the increased IFNγ after immunisation with *Nm* PorB OMVs suggests that *Nm* PorB OMVs prime a Th1-biased response.

IL-4 is an effector cytokine indicative of a Th2 response [54]. There was a trend to higher IL-4 production by splenocytes from *Nm* PorB OMV immunised mice compared to *Ng* PorB OMV immunised mice when stimulated with either *Ng* PorB OMVs (mean 13.3 pg/mL compared to 7.7 pg/mL) or *Nm* PorB OMVs (mean 14.2 pg/mL compared to 9.5 pg/mL) although the differences were not statistically significant (Fig 8B). Splenocytes from both the *Ng* PorB OMV and *Nm* PorB OMV immunised groups produced significantly more IL-4 when re-stimulated with OMVs when compared to the PBS immunised control group ($p < 0.0001$). Together this suggests that immunisation with *Ng* PorB OMVs or *Nm* PorB OMVs primes Th2 responses to a similar extent. The production of cytokine IL-10, involved in modulating the inflammatory response, was significantly increased only in splenocytes from mice immunised with *Nm* PorB OMV when re-stimulated with *Nm* PorB OMVs ($p < 0.05$, Fig 8C).

As gonococcal infection drives a Th17 response [23], we examined the production of IL-17A as an indicator of Th17 responses. Splenocytes from both the *Ng* PorB OMV and *Nm* PorB OMV immunised groups produced significantly more IL-17A when re-stimulated with OMVs when compared to the PBS immunised control group ($p < 0.001$, Fig 8D). However, there was no significant difference in IL-17A production between the groups of mice immunised with *Ng* PorB OMVs or *Nm* PorB OMVs (Fig 8D), suggesting that changing PorB did not impact the murine Th17 response.

IL-6 is a pleiotropic cytokine that participates in both pro- and anti-inflammatory responses. After stimulation with *Ng* PorB OMVs, IL-6 production was similar in murine splenocytes from naïve mice and mice immunised with *Ng* PorB OMVs. However, a difference was observed between these groups when stimulated with *Nm* PorB OMVs, with the latter group producing more IL-6 ($p < 0.05$). IL-6 production was also significantly higher by splenocytes from *Nm* PorB OMV immunised mice when compared to the *Ng* PorB OMV immunised group, when re-stimulated with either *Ng* PorB OMVs ($p < 0.0001$) or *Nm* PorB OMVs ($p < 0.0001$) (Fig 8E). IL-2 is produced by T cells and is also a pleiotropic cytokine that does not promote any particular Th-response, shown to positively influence the differentiation, expansion, and maintenance of both Th1- and Th2-type cells, as well as T regulatory (Treg) cells and effector T cells [55,56]. Production of IL-2 was significantly higher in murine splenocytes stimulated with OMVs after immunisation with either *Ng* PorB OMVs or *Nm* PorB OMVs when compared to naïve splenocytes ($p < 0.001$), suggesting an antigen-specific expansion of T cells in response to immunisation with OMVs. However, there was no significant difference in IL-2 production between the groups of mice immunised with *Ng* PorB OMVs or *Nm* PorB OMVs (Fig 8F).

To summarise, immunisation with *Ng* or *Nm* PorB OMVs elicits antigen-specific T cell expansion in splenocytes after re-stimulation with OMVs, evidenced by increased IL-2 production from splenocytes compared to control groups. Splenocytes from both *Ng* and *Nm* PorB OMV immunised mice elicited similar production of IL-4, suggesting a similar level of Th2 response. However, *Nm* PorB OMVs elicited a more pro-inflammatory response, producing significantly more IFNγ, indicative of a Th1-biased response, which is beneficial for protection against gonococcal infection [24,57]. Potentially related to high IFNγ production, both regulatory cytokines IL-6 and IL-10 were significantly higher when splenocytes from *Nm* PorB OMV immunised mice were re-stimulated with *Nm* PorB OMVs.

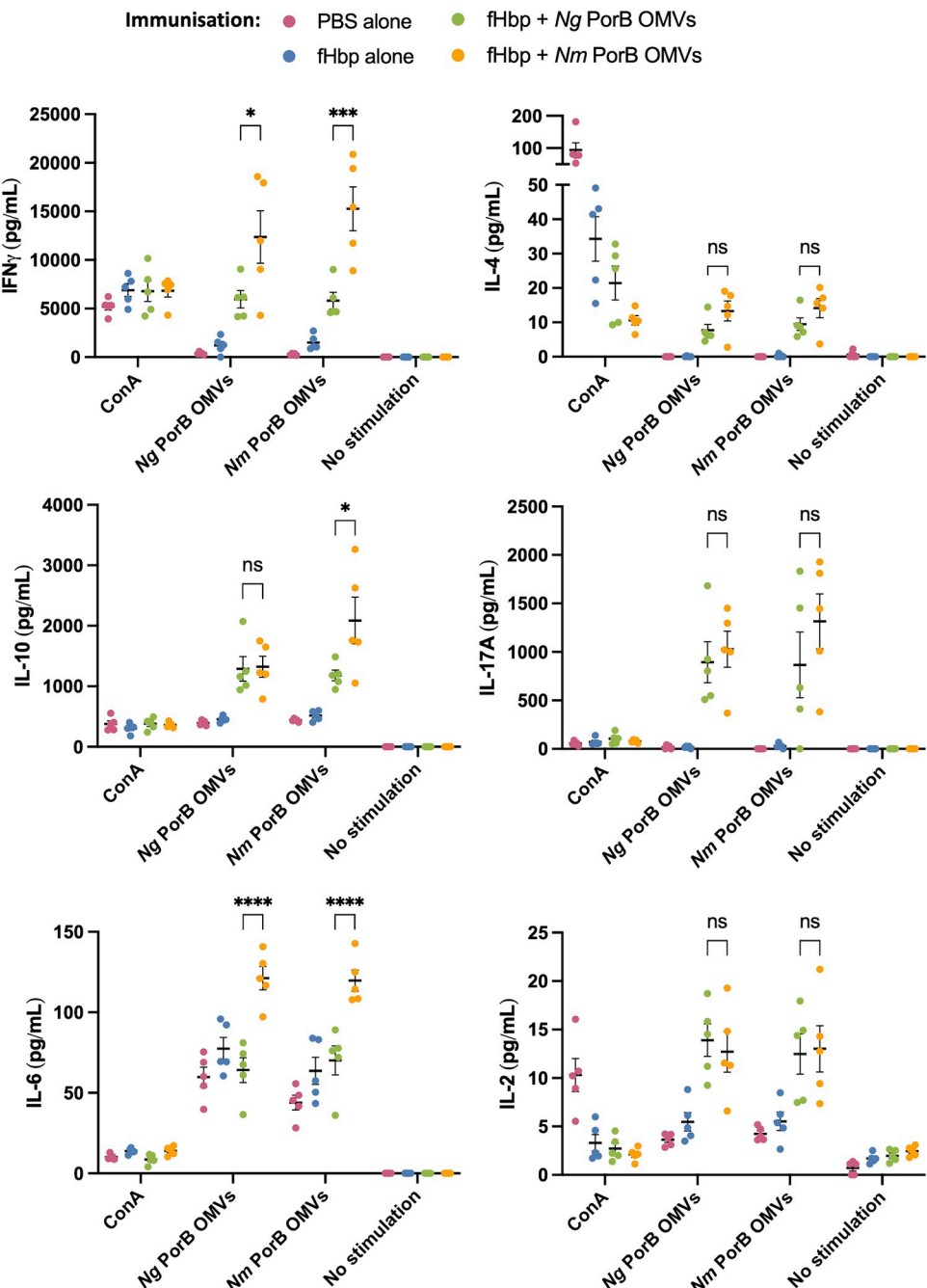

**Fig 8. Cytokines produced by murine splenocytes after immunisation with *Ng*- or *Nm*-PorB OMVs, when re-stimulated with concanavalin A (ConA, positive control), *Ng* PorB OMVs, *Nm* PorB OMVs or no stimulation.** A) interferon-γ (IFNγ), B-F) interleukin (IL)-4, IL-10, IL-17A, IL-6 and IL-2. Data are the mean ± standard error of the mean. * $p < 0.05$, *** $p < 0.001$, **** $p < 0.0001$, ns: not significant.

## Discussion

As the incidence and treatment failures due to multi-drug resistance in *N. gonorrhoeae* are increasing, there is a significant global public health need to develop a successful vaccine against the gonococcus. The current status of gonococcal vaccine research has been recently

reviewed [58]. For OMV-based vaccines, research has focused on improving *N. meningitidis* OMVs for use against the gonococcus [59,60]. This approach is informed by retrospective analyses and randomised control trials that demonstrate cross-protection against gonococcal infection after immunisation with meningococcal OMV-based vaccines [17–19,21]. While this approach shows promise, it is obviously ideal that a vaccine against the gonococcus is derived directly from antigens from the gonococcus itself rather than another species. Three *N. gonorrhoeae*-specific OMV vaccines are in late pre-clinical or early clinical stages [58]. However, to our knowledge, deletion of *rmpM* is the only approach used in these OMV vaccines to address the issue of gonococcal manipulation of host immune responses. Here we demonstrate that the presence of immunomodulatory gonococcal PorB [30], the most abundant protein in OMVs, significantly suppresses immune responses, potentially impairing the success of gonococcal vaccines that utilise *N. gonorrhoeae*-derived OMVs.

PorB is essential for the viability of *N. gonorrhoeae*. Therefore, we decided to replace the *porB* gene in *N. gonorrhoeae* FA1090Δ*rmpM* with *porB* from *N. meningitidis* MC58. Gonococcal PorB has numerous immunomodulatory properties, whereas meningococcal PorB has adjuvant properties that promote immune responses [30,40–42]. The exact reasons for the differences in properties between pathogenic *Neisseria* PorB proteins are unknown. The effect of changing PorB on humoral responses was assessed by immunising mice with *Nm*/*Ng* PorB OMVs alongside the model antigen fHbp. The presence of *Nm* PorB OMVs led to higher antibody titres against fHbp when compared to *Ng* PorB OMVs. Higher antibody titres were also found against MetQ and MtrE, antigens in the OMVs. The increased antibody responses to MetQ and MtrE could be due to their higher abundances in *Nm* PorB OMVs compared to *Ng* PorB OMVs (abundance ratios (*Nm*/*Ng*) of 6.1 and 2.2, respectively). However, immunisation with *Nm* PorB OMVs elicited greater murine antibody reactivity against a range of gonococcal antigens, shown by using protein microarrays. This included stronger reactivity against SliC, which has a similar abundance in *Ng* and *Nm* PorB OMVs (abundance ratio of 1.1), and also BamE, which is actually more abundant in *Ng* PorB OMVs (abundance ratio of 0.07). MtrE, MetQ and SliC have shown promise as vaccine candidates [24,25,61]. Therefore, stronger antibody responses against these candidate antigens is a positive attribute of *Nm* PorB OMVs as a gonococcal vaccine. Furthermore, transcriptomic data from infected patients shows that bacteria express *mtrE*, *metQ* and *sliC* [62,63].

Mass spectrometry demonstrated that replacing *porB* in *N. gonorrhoeae* FA1090 resulted in additional differences in the proteomes of *Ng* PorB OMVs and *Nm* PorB OMVs, other than PorB itself. The additional changes included proteins related to nutrient acquisition and host-pathogen interactions, which may be beneficial for a vaccine as these proteins are important during infection. The increased expression of nutrient acquisition proteins could relate to the growth defect of *N. gonorrhoeae* expressing meningococcal PorB. Of the proteins more highly represented in *Nm* PorB OMVs, there are no obvious candidates that could have a large impact on the enhanced immune responses we observed. However, we cannot rule out the possibility that these proteins contribute currently unknown immunomodulatory properties. As Opa proteins are also linked to the suppression of Th1/Th2 responses [23], it is unlikely that the difference in the Opa profiles of the OMVs is responsible for the enhanced immune responses. Further studies are required to determine whether the change from *Ng* to *Nm* PorB consistently influences Opa profiles in independent mutants. Absolute quantification of PorB showed that the porin represented a significantly higher proportion of *Ng* PorB OMVs compared to *Nm* PorB OMVs, contributing 8.8 μg of protein compared to 4.6 μg of protein per immunisation dose, respectively. Previous data demonstrate that inhibition of T cell proliferation by gonococcal PorB is concentration dependent [64]. Therefore, the relative reduction in PorB in *Nm* PorB OMVs could also contribute to the enhanced immune responses through

reduced suppression of the adaptive immune responses. It remains to be determined whether simply decreasing the expression level of native gonococcal PorB would confer similar benefits.

Recent data has indicated that driving a Th1-mediated response could be important for the success of a gonococcal vaccine [9, 24, 57, 65]. Indirect analysis of Th1/Th2 responses through determining IgG1/IgG2a ratios showed that immunisation with *Nm* PorB OMVs promoted higher IgG2a antibody titres against OMV antigens MetQ and MtrE, compared to *Ng* PorB OMVs, indicating a Th1 driven response. Interestingly, *Nm* PorB OMVs resulted in a strong Th2 response against recombinant protein fHbp, suggesting that the method of antigen presentation (*i.e.* as a recombinant protein rather than present in OMVs) influences the type of immune response. Increased immunogenicity of recombinant proteins after addition of OMVs was noted during the development of meningococcal B vaccines, shown by increased serum bactericidal activity against *N. meningitidis* [66,67]. Given that murine IgG2a is more potent in bactericidal activity than IgG1 [68], we were surprised to find similar SBA titres against *N. gonorrhoeae* FA1090, G97687 and 60755 in sera after *Ng* or *Nm* PorB OMV immunisation. G97687 is a multidrug resistant strain, while 60755 was isolated in Kenya, and belongs to a lineage of *N. gonorrhoeae* that had not been identified previously [48,49]. The specific epitope(s) that confer bactericidal activity remain to be determined. Future studies would benefit from assessing opsonophagocytic activity to further characterise the antibody response. Intriguingly, antibodies from mice immunised with *Nm* PorB OMVs had very little cross-reactivity to *N. gonorrhoeae* FA1090 PorB included on our protein microarray, perhaps suggesting that antibody responses to PorB are not a contributing factor to the cross-protection elicited by Bexsero.

The Th1 response elicited by *Nm* PorB OMVs is further evidenced by higher IFNγ production by murine splenocytes after re-stimulation with either *Ng* or *Nm* PorB OMVs, when compared to splenocytes from *Ng* PorB OMV immunised mice. IL-4 was measured as an indicator of Th2 responses, where splenocytes from *Nm* PorB OMV immunised mice exhibited a non-significant trend to higher IL-4 production compared to *Ng* PorB OMV immunised mice. In mice, IL-6 plays a key role in antigen-specific IgG production [69, 70], therefore the significant IL-6 response in splenocytes after *Nm* PorB OMV immunisation may explain the increase in IgG antibody titres against antigens present in the immunisation formula. Overall, immunisation with *Ng* PorB OMVs did promote some Th1/Th2 cellular differentiation, but not to the same extent as immunisation with *Nm* PorB OMVs, which is consistent with gonococcal suppression of Th1/Th2 responses [23] and suggests that the change in PorB contributed to improving murine cellular immune responses.

In summary, replacing gonococcal PorB with meningococcal PorB in *N. gonorrhoeae*-derived OMVs represents a promising gonococcal vaccine platform, eliciting enhanced murine humoral and cellular immune responses, including an important Th1-skew. In the present study, we focus on tackling gonococcal immunosuppression to improve murine immune responses to OMVs. As gonococcal OMVs alone have proved unsuccessful as a vaccine in the past [71,72], and have shown more promise alongside Th1-priming adjuvants such as microencapsulated IL-12 [65], further efforts are required to explore further additions to the *Nm* PorB OMV platform before assessing protective efficacy *in vivo*. For example, we did not include LOS detoxification, often used to reduce the reactogenicity of OMVs, through detergent extraction of OMVs or genetic alterations such as deletion of *lpxl1*. A key question remains as to whether LOS detoxification would impact the immune responses to *Nm* PorB OMVs that we have observed. Equally, detergent extraction of OMVs can result in loss of important lipoproteins, for example fHbp from meningococcal OMVs [73], and could potentially remove key gonococcal lipoproteins. Therefore, the mode of preparation of OMVs is an

important future consideration. However, LOS detoxification could allow *Nm* PorB OMVs to be combined with different adjuvants that could further optimise the immune response, and could be particularly important for influencing the immune response to recombinant proteins added to, and not incorporated in, the OMVs. This would allow the addition of key recombinant antigens alongside *Nm* PorB OMVs, that together drive a strong Th1 response with the aim of protecting from gonococcal infection.

## Materials and methods

### Ethics statement

All animal experiments were carried out under protocols reviewed by the Animal Welfare and Ethical Review Board at the Sir William Dunn School of Pathology, and approved by the Home Office, UK, under licence number PPL P20CC6E82.

### Bacterial growth and transformation

*N. gonorrhoeae* strain FA1090 was grown on gonococcal base medium (GCB) agar plates or in liquid media containing 1% Vitox at 37˚C in 5% $CO_2$. Mutants were generated by spotting DNA fragments onto GCB plates, streaking *N. gonorrhoeae* FA1090 over the spot, and incubating at 37˚C and 5% $CO_2$ for 8 hours. Growth over the spot was then plated onto selective media (kanamycin at 80 μg/mL or erythromycin at 2 μg/mL, as appropriate) and incubated overnight at 37˚C in 5% $CO_2$. Mutants were confirmed by PCR and Sanger sequencing.

### Genetic manipulation of *N. gonorrhoeae*

Genomic DNA from *N. meningitidis* strain MC58 was used as the template to amplify *N. meningitidis porB* (primer pair: MC58 *porB* F and R, all primers are listed in S1 Table), and to add downstream complementary sequences to a kanamycin resistance cassette *aph(3)* [74] (*kanR*). DNA upstream (799 bp, primers FA1090 *porB* upstream F and R) and downstream (690 bp, primers FA1090 *porB* downstream F and R) of *porB* was amplified from FA1090 genomic DNA by PCR adding DNA sequences complementary to *N. meningitidis* MC58 *porB* and *kanR*; Q5 polymerase was used throughout (New England Biolabs),. PCR products were fused by Gibson assembly (New England Biolabs), and the final construct was re-amplified and purified by gel extraction (Promega Wizard) before being used to transform *N. gonorrhoeae* FA1090. Insertion of the *N. meningitidis porB* gene was confirmed by PCR, Sanger sequencing, and Western blot analysis, with the resultant strain named FA1090*Nm porB*.

The *ermC'* erythromycin resistance cassette [74] (referred to as *eryR*) was amplified by PCR, with the addition of sequences complementary to regions upstream and downstream of *N. gonorrhoeae* FA1090 *rmpM* (primer pair: *eryR* F and R). DNA sequences downstream (620 bp, primer pair: *rmpM* downstream F and R) and upstream (578 bp, primer pair: *rmpM* upstream F and R) of *rmpM* were amplified from FA1090 genomic DNA. The three PCR products were fused by Gibson assembly (New England Biolabs) and used to transform *N. gonorrhoeae* FA1090 and FA1090*Nm porB*. Deletion of *rmpM* was confirmed by PCR, Sanger sequencing and Western blot, and the resultant strains named FA1090ΔrmpM and FA1090*Nm porB*Δ*rmpM*. The genomic sequences upstream and downstream of *N. gonorrhoeae* FA1090 *lgtE* were amplified by PCR (primer pairs: *lgtE* upstream F and *lgtE* upstream R; *lgtE* downstream F and *lgtE* downstream R) with the addition of complementary sequences to the *kanR* resistance cassette. PCR products were fused by Gibson assembly and used to transform *N. gonorrhoeae* FA1090 as before, and the mutant strain confirmed by Sanger sequencing.

## Production of OMVs

Bacteria were grown overnight on GCB agar plates then resuspended in 50 mL liquid GCB at an optical density ($OD_{600}$) of 0.1 for FA1090$\Delta rmpM$ and 0.2 for FA1090$_{Nm\ porB}$ $\Delta rmpM$, and incubated at 37˚C in 5% $CO_2$ with shaking at 180 rpm. Upon reaching an $OD_{600}$ of 1, bacteria were pelleted by centrifugation at 2,739 x $g$ for 20 minutes at 4˚C. Supernatants were passed through 0.22 µm pore size filters, and OMVs obtained *via* ultracentrifugation of the supernatant at 235,000 x $g$ for 2 hours at 4˚C. OMVs were washed with PBS before ultracentrifugation as before, then stored at 4˚C. The protein content of OMV preparations was determined by BCA assay (Pierce).

## Protein purification

*mtrE* and *metQ* were amplified from *N. gonorrhoeae* FA1090 and ligated into pET14b, generating pET14b:*mtrE and* pET14b:*metQ*, respectively. For protein expression, *E. coli* B834 containing pET14b:*mtrE* or pET14b:*metQ* was grown in TB medium for 24 h at 22˚C and harvested by centrifugation at 5,000 × $g$ for 30 minutes at 4˚C, prior to cell lysis using an EmulsiFlex-C5 homogeniser (Avestin, 15,000 lb/in$^2$). Lysates were centrifuged at 20,000 × $g$ for 30 minutes at 4˚C, and recombinant protein was bound to HisTrap columns (GE Healthcare), eluted with 300 mM imidazole, and further purified by size exclusion (AKTA, HiLoad 16/600 Superdex 200 pg column; GE Healthcare). Protein concentrations were estimated using a Nanodrop 2000c spectrophotometer (Thermo Scientific). Factor H binding protein (fHbp) v1.1 was expressed in *E. coli* B834 during growth at 22˚C for 24 hours with 1 mM IPTG (final concentration). Bacteria were harvested and resuspended in Buffer A (50 mM Na-phosphate pH 8.0, 300 mM NaCl, 30 mM imidazole) and fHbp purified by Nickel affinity chromatography (Chelating Sepharose Fast Flow; GE Healthcare). Columns were washed with Buffer A, then with 80:20 Buffer A:Buffer B (50 mM Na-phosphate pH 8.0, 300 mM NaCl, 300 mM imidazole) and proteins eluted in 40:60 Buffer A:Buffer B. Proteins were dialysed overnight at 4˚C into PBS, 1 mM DTT pH 8.0 with TEV protease prior to Nickel affinity chromatography to remove the HIS-GST-TEV. fHbp was eluted from Sepharose columns with Buffer B after washing with buffer C (50 mM Na-phosphate pH 6.0, 500 mM NaCl, 30 mM imidazole), and dialysed overnight at 4˚C into Tris pH 8.0. Proteins were filtered through a 0.22 µm pore size filter before use in immunisations.

## Immunisation of mice with OMVs

Six-week-old female BALB/c mice (five per group) were immunised by the intraperitoneal route with: i) PBS alone, ii) 10 µg fHbp alone, iii) 10 µg fHbp with 12.5 µg FA1090$\Delta rmpM$ OMVs, or iv) 10 µg fHbp with 12.5 µg FA1090$_{Nm\ porB}$ $\Delta rmpM$ OMVs. All components were diluted in PBS (200 µL total volume per dose). Mice received three immunisations (on days 1, 21 and 35) before sera was obtained for analysis on day 49.

## Antibody titre determination by ELISA

Microplates (Nunc Immuno Maxisorp, Thermo Scientific) were coated with the target recombinant protein (fHbp v1.1, MtrE or MetQ) by adding 50 µL of 2.5 µg/mL protein diluted in PBS and incubating overnight at 4˚C. PBS was added to control wells. Plates were washed three times with PBS + 0.5% Tween20 (PBS-T) before blocking with 4% bovine serum albumin (BSA) in PBS-T for 1 hour at 37˚C. Plates were washed as before, before addition of pooled sera from immunisation groups in threefold serial dilutions, and incubated for 1 hour at 37˚C. Plates were washed as before, and anti-mouse IgG-HRP (Bio-Rad, 172–1011) conjugated

secondary antibody was added at a 1:10,000 dilution and incubated for 1 hour at 37˚C. Alternatively, anti-mouse IgG1-HRP (Abcam, ab97240) or IgG2a-HRP (Abcam, ab97245) were added at a 1:20,000 dilution, or anti-mouse IgG1-AF488 (Invitrogen, A21121) or IgG2a-AF488 (Invitrogen, A21131) were added at a 1:1000 dilution. After further washes, plates were developed by adding TMB ELISA substrate followed by stop solution (BioTechne), and the absorbance read at 450 nm in a FLUOStar Omega plate reader (BMG LabTech). For AF488 conjugated secondary antibodies, fluorescence was measured using 485/520 nm for excitation/ emission, respectively. To calculate endpoint titres, the baseline value was set at three times the average of the PBS controls. The endpoint titre is reported as the $\log_{10}$ of the reciprocal of the serum dilution that was equal to the baseline value. Statistical significance was tested by two-way ANOVA with multiple comparisons in GraphPad Prism (GraphPad Software Inc. v.10.0).

## Flow cytometry

*N. gonorrhoeae* strains FA1090, G97687 and 60755 were grown overnight on GCB agar plates then resuspended in 5 mL liquid GCB at an $OD_{600}$ of 0.1 for FA1090 and 0.2 for G97687 and 60755, and incubated at 37˚C in 5% $CO_2$ with shaking at 180 rpm. Upon reaching an $OD_{600}$ of 0.8, 1 mL of the bacteria suspension was pelleted by centrifugation at 16,000 x *g* for 5 minutes, washed once with supplemented PBS, and 100 μL bacteria suspension was incubated in a two-fold serial dilution of murine sera in supplemented PBS with 1% BSA (w/v) (1:500 to 1:16,000) for 30 minutes at RT with shaking at 1250 rpm. Bacteria were washed twice as before, followed by incubation with goat anti-mouse IgG-Alexa fluor 647 secondary antibody (Invitrogen) at a 1:1000 dilution for 30 minutes at RT with shaking at 1250 rpm. Bacteria were washed twice as before and fixed using 3% paraformaldehyde before analysing on flow cytometer BD LSRFortessa X-20. FlowJo v10.10 software was used to calculate the mean fluorescence intensity for each spectra.

## Lipooligosaccharide analyses

*N. gonorrhoeae* WT FA1090 and FA1090Δ*lgtE* were grown overnight on GCB agar plates then resuspended in PBS at an $OD_{600}$ of 1 and 750 μL of bacterial suspension was pelleted by centrifugation at 16,000 x *g* for five minutes before resuspending in 200 μL PBS. Bacterial suspensions or 15 μg of OMV preparations were digested with proteinase K (final concentration 1 mg/mL) at 59˚C for a minimum of six hours. LOS resulting from proteinase K digested OMVs or bacterial suspension was separated on a 16% tricine gel, as previously described [75], and transferred onto PVDF membrane (0.22 μm, Immobilon-PQS) at 15 V for 90 minutes using a semi-dry blot (Bio-Rad). Membranes were blocked overnight at 4˚C in 5% skim milk in PBS-T before probing with either an anti-2C7 monoclonal antibody (antibody registry ID: AB_2889813, Developmental Studies Hybridoma Bank, University of Iowa) [51] at a final concentration of 0.25 μg/mL or pooled sera from OMV-immunised mice at a 1:2000 dilution. After three washes in PBS-T, goat anti-mouse IgG-HRP (Bio-Rad) was added at a 1:10,000 dilution. After final washes, membranes were developed using Amersham ECL Western Blotting Analysis System (GE Healthcare) and exposed to Amersham Hyperfilm ECL (GE Healthcare). For ELISA analysis, microplates (Nunc Immuno Maxisorp, Thermo Scientific) were coated with LOS at a protein equivalent of 2 μg of proteinase K digested WT FA1090 overnight at 4˚C. After blocking with 4% BSA in PBS-T, 100 μL anti-2C7 mAb at 0.5 μg/mL was added to appropriate wells prior to incubation with serial dilutions of OMV-immunised murine sera. Anti-mouse IgG1-HRP (Abcam, ab97240) and IgG2a-HRP (Abcam, ab97245) were added together at a 1:10,000 dilution, and plates were developed by adding TMB ELISA substrate followed by stop solution (BioTechne), and the absorbance read at 450 nm in a FLUOStar

Omega plate reader (BMG LabTech). Wells with LOS and 2C7 mAb alone and wells with secondary antibody only were used as background controls.

## SDS-PAGE gel electrophoresis and Western blotting

*N. gonorrhoeae* cell lysates (10 μL) or OMVs (5 μg) were separated on 12% SDS-polyacrylamide gels. SDS-PAGE gels were either Coomassie stained, or proteins were transferred to 0.45 μm nitrocellulose membranes, using a semi-dry blot at 25 V for 30 minutes (Bio-Rad). Membranes were blocked overnight in 5% milk in PBS-T at 4°C, before the addition of primary antibody and incubation at room temperature for 2 hours. Primary antibodies were diluted in 5% milk in PBS-T at 1:1000 for anti-PorB antibodies (P3.15 for meningococcal PorB and H5.2 for gonococcal PorB, NIBSC, UK) and 1:10,000 for the anti-RmpM antibody [76]. After three washes in PBS-T, membranes were incubated with polyclonal goat anti-mouse immunoglobulins secondary antibody conjugated to HRP (Dako, P0447), diluted 1:10,000. After final washes, membranes were developed using Amersham ECL Western Blotting Analysis System (GE Healthcare) and exposed to Amersham Hyperfilm ECL (GE Healthcare).

## Splenocyte re-stimulation and cytokine quantification

Two weeks after the final immunisation, mouse spleens (five per group) were harvested and splenocytes were pressed through a 70 μm cell strainer, and suspended in RPMI-1640 supplemented with 10% Foetal Bovine Serum (FBS) and 50 μM 2-mercaptoethanol (complete RPMI). The cell suspension was pelleted by centrifugation at 300 x *g* for 10 minutes, and the pellet was resuspended in 3 mL ACK lysis buffer for 5 minutes to remove red blood cells. Cells were washed with 10 mL complete RPMI, centrifuged as before, resuspended in PBS and counted using a haemocytometer. Splenocytes were diluted to $2 \times 10^6$ cells/mL in complete RPMI and 1 mL was plated per well in a 24-well plate. Splenocytes were then stimulated with 2 μL per mL concanavalin A (500X stock, eBioscience), 5 μg *Ng* PorB OMVs, 5 μg *Nm* PorB OMVs, or left unstimulated, and incubated for 72 hours at 37°C with 5% $CO_2$. Cell culture supernatants were stored at -80°C until analysis. The release of cytokines interferon-gamma, IL-2, IL-4, IL-6, IL-10 and IL-17A were quantified by ELISA (Abcam) according to the manufacturer's instructions and diluting the supernatants as appropriate. Statistical significance was tested by two-way ANOVA with multiple comparisons in GraphPad Prism (GraphPad Software Inc. v.10.0).

## Serum bactericidal assay

*N. gonorrhoeae* was grown overnight on GCB agar and colonies were resuspended in PBS to an $OD_{600}$ of 0.07 (~ 6–7 x $10^3$ colony forming units (CFU)/mL) and incubated with serial dilutions of pooled from five mice heat-inactivated (56°C for 1 hour) murine sera for 10 minutes at 37°C with 5% $CO_2$. IgG-depleted human sera (Pel-freeze) was then added at a final concentration of 3% (v/v) for strain FA1090 and 10% (v/v) for clinical isolates G97687 and 60755 [48, 49]. and incubated at 37°C with 5% $CO_2$ for 45 minutes. The dilutions were plated on GCB agar in triplicate as 10 μL spots and incubated for 20–28 hours at 37°C with 5% $CO_2$. Survival was calculated as a percentage of the colonies present in the IgG-depleted human sera only control.

## Mass spectrometry analysis of OMVs

OMVs were subjected to trypsin digestion before separation and mass spectrometric (MS) analysis of the tryptic peptides was performed on an Vanquish Neo UHPLC coupled to

Orbitrap Eclipse Tribrid Mass Spectrometer (Thermo Fisher Scientific). For LC-MS/MS analysis, 4 μL of tryptic sample was injected. Samples were further purified and concentrated on a trap column (C18 PepMap, 300 μm ID x 5 mm, 5 μm, 100 Å; Thermo Fisher Scientific), then separated on a 15 cm analytical nano-LC column (PepMap C18, 75 μm ID x 150 mm, 2 μm, 100 Å; Thermo Fisher Scientific) using a binary 60-minute linear gradient from 5% to 45% buffer B (80% acetonitrile (ACN), 0.1% formic acid (FA)) for 60 minutes, then to 99% buffer B over 1 minute and maintained at 99% for a further 10 minutes, at a flow rate of 300 nL/min. Solvent A is a buffer containing 2% ACN and 0.1% FA in water.

Orbitrap Eclipse Tribrid Mass Spectrometer equipped with a Thermo Easy-Spray capillary Emitter was used to analyse the separated peptides. Protein identification and label-free quantification (LFQ) were achieved using Proteome Discoverer (PD; version 2.5, Thermo Fisher Scientific). For protein identification, raw files were searched against a database containing canonical protein sequences of the *N. meningitidis* MC58 PorB and *N. gonorrhoeae* FA1090 (https://www.uniprot.org) with the Sequest HT search algorithm and using modified standard processing and consensus workflows. The false discovery rate (FDR) tolerance in the Percolator node was set to 0.01 for high confidence and 0.05 for medium confidence.

Absolute quantification of PorB in OMVs was carried out using the same method developed previously for quantifying porins in Bexsero [77]. Briefly, PorB peptide standards, a native and a heavy (isotopically labelled) analogue, were synthesised by Thermo Scientific (HeavyPeptide AQUA Ultimate service, 97% purity; 99% isotope enrichment). The heavy peptides were $^{13}C^{15}N$ labelled; for SDYLGVNK, the valine residue was labelled, for GQEDLGNGLK, the penultimate leucine was labelled. Labelled peptide was added to a final concentration of 70 fmol/μL to the tryptic digests of OMVs before separation by LC/MS. Peak area ratios of tryptic-released native peptide/heavy peptide were compared with a PorB peptide standard curve [78] for each peptide to provide a surrogate estimate of the molar content of PorB in the digest.

## Gonococcal protein microarray

The gonococcal microarray slides were custom-made by Arrayjet Limited, UK, as described previously [79]. Each microarray slide consisted of 16 identical blocks, with each mini-array containing 91 individually purified gonococcal proteins (derived from *N. gonorrhoeae* strain FA1090) and 17 control samples, printed in five repeats. Three sets of control samples were used for the slides, consisting of monoclonal mouse IgG, human IgG and EBNA-1 viral protein. Mouse and human IgGs were prepared at eight concentrations, with the highest concentration of 0.5 mg/mL and serially diluted with JetStar printing buffer (Arrayjet, UK) down to 0.0035 mg/mL. EBNA-1 viral protein at 0.05 mg/mL concentration was also included as a negative control. The slides were stored at 4˚C until use.

## Microarray immunogenicity probing and data acquisition

The slides were blocked with 3 mL of SuperG Blocking Buffer (Grace Biolabs, US) per slide and incubated for one hour at room temperature (RT). Sera were diluted 1:300 with TBS, and 50 μL was added to each mini-array and incubated at 20˚C for one hour. After washing three times with 300 μL TBS-T (0.05% Tween-20) and once in TBS for 10 minutes, slides were incubated in the dark for 1 hour at 20˚C with 50 μL of goat anti-mouse IgG Fc (DyLight 650) (ab97018, Abcam, UK) diluted 1:5000 in blocking agent. The same protocol was followed for determining IgG1 and IgG2a, using a goat anti-mouse IgG1 cross-adsorbed secondary antibody (Alexa Fluor 647) (A-21240, Invitrogen) and a goat anti-mouse IgG2a cross-adsorbed secondary antibody (Alexa Fluor 488) (A-21131, Invitrogen). Slides were rinsed in de-ionised

water and dried by centrifugation at 200 x *g* for 2 minutes. Slides were scanned in an InnoScan 710 (Innopsys, France) with the photomultiplier tube set to 40% for 635 nm. Image analysis and data quantification were carried out using Mapix—Microarray image acquisition and analysis software (v9.1.0, Innopsys, France). Microarray spot intensities were quantified using automatic background subtraction for each spot. The spot intensities for each protein were recorded in quintuplicate; arithmetic means were determined, and spot intensities for buffer-only controls were subtracted.

## Supporting information

**S1 Data. Raw data used to plot endpoint ELISA titres and flow cytometry median fluorescence intensity in Fig 3.**
(XLSX)

**S2 Data. Raw data used to plot ELISA graph for serum antibody binding to gonococcal LOS with and without anti-2C7 blocking in Fig 4.**
(XLSX)

**S3 Data. Raw data used to plot IgG heatmap (Fig 5) and volcano plots (S5 Fig).**
(XLSX)

**S4 Data. Raw data used to plot IgG1 and IgG2a heatmaps in Fig 6.**
(XLSX)

**S5 Data. Raw data used to plot each cytokine production graph in Fig 8.**
(XLSX)

**S6 Data. Raw data used to plot growth curves in S2 Fig.**
(XLSX)

**S7 Data. Raw data used to plot SBA graphs in S4 Fig and to calculate SBA titres in Table 1.**
(XLSX)

**S1 Table. List of primers used in this study.**
(DOCX)

**S2 Table. All proteins identified by mass spectrometry in *Ng* and *Nm* PorB OMVs.** Label-free quantification of three independent replicates of each OMV.
(XLSX)

**S3 Table. Proteins identified by mass spectrometry in *Ng* and *Nm* PorB OMVs with specific conditions applied.** Filtered protein list that removed proteins with low coverage, low peptide number or proteins only identified in one sample.
(XLSX)

**S1 Fig. Amino acid sequence alignment of PorB from *N. gonorrhoeae* FA1090 and *N. meningitidis* MC58. Gaps are represented by '-'. '*' indicates identical amino acids, ':' indicates conservative substitution and '.' indicates semi-conservative substitution.**
(TIFF)

**S2 Fig. Growth of *N. gonorrhoeae* FA1090 and mutant strains expressing *N. meningitidis* MC58 *porB* (*Nm porB*), with or without deletion of *rmpM*.**
(TIFF)

**S3 Fig. Coomassie-stained 12% SDS-PAGE separation of three independent replicates of outer membrane vesicles (OMVs) obtained from *N. gonorrhoeae* FA1090 strains as**

indicated Δ*rmpM* and FA1090~MC58 PorB~ Δ*rmpM*. *Ng* PorB is indicated ~35 kDa and *Nm* PorB at 37 kDa.
(TIFF)

**S4 Fig.** Serum bactericidal activity against WT *N. gonorrhoeae* FA1090 (A), G97687 (B) and 60755 (C) after immunisation with *Ng*- or *Nm*-PorB OMVs. NHS: normal human serum (IgG and IgM depleted), HI: heat inactivated.
(TIFF)

**S5 Fig.** Volcano plots of total IgG responses against all antigens in the microarray A) fHbp versus *Nm* PorB OMVs + fHbp B) fHbp versus *Ng* PorB OMV + fHbp. One sided t-tests were conducted to compare IgG responses for each antigen in each paired group. The negative log10 of each p-value is plotted against log2 (binary) of the mean fold-change i.e. (mean IgG response group 2)/(mean IgG response group 1). Antigens with -log10(p-value) > 4 are labelled.
(TIFF)

## Author Contributions

**Conceptualization:** Rebekah A. Jones, Ann E. Jerse, Ana Cehovin, Christoph M. Tang.

**Data curation:** Rebekah A. Jones, Fidel Ramirez-Bencomo, Gail Whiting, Min Fang, Kacper Kurzyp, Angela Thistlethwaite, Jeremy P. Derrick.

**Formal analysis:** Rebekah A. Jones, Fidel Ramirez-Bencomo, Gail Whiting, Min Fang, Angela Thistlethwaite, Jeremy P. Derrick.

**Funding acquisition:** Christoph M. Tang.

**Investigation:** Rebekah A. Jones, Fidel Ramirez-Bencomo, Gail Whiting, Min Fang, Hayley Lavender, Ann E. Jerse, Ana Cehovin, Jeremy P. Derrick.

**Methodology:** Rebekah A. Jones, Fidel Ramirez-Bencomo, Gail Whiting, Min Fang, Hayley Lavender, Kacper Kurzyp, Lenka Stejskal, Smruti Rashmi, Ana Cehovin, Jeremy P. Derrick.

**Project administration:** Christoph M. Tang.

**Resources:** Hayley Lavender, Kacper Kurzyp, Angela Thistlethwaite, Lenka Stejskal, Smruti Rashmi, Ana Cehovin.

**Supervision:** Gail Whiting, Ann E. Jerse, Ana Cehovin, Jeremy P. Derrick, Christoph M. Tang.

**Validation:** Rebekah A. Jones, Fidel Ramirez-Bencomo, Gail Whiting, Min Fang, Christoph M. Tang.

**Visualization:** Rebekah A. Jones, Fidel Ramirez-Bencomo, Gail Whiting, Min Fang, Jeremy P. Derrick, Christoph M. Tang.

**Writing – original draft:** Rebekah A. Jones, Jeremy P. Derrick, Christoph M. Tang.

**Writing – review & editing:** Rebekah A. Jones, Fidel Ramirez-Bencomo, Gail Whiting, Min Fang, Hayley Lavender, Kacper Kurzyp, Angela Thistlethwaite, Lenka Stejskal, Smruti Rashmi, Ann E. Jerse, Ana Cehovin, Jeremy P. Derrick, Christoph M. Tang.

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
