## [Decision Letter · Decision Letter 0]

3 Jul 2024

Dear Professor Tang,

Thank you very much for submitting your manuscript "Tackling immunosuppression by Neisseria gonorrhoeae to facilitate vaccine design" for consideration at PLOS Pathogens.

We are sincerely sorry for the unusual delay in finding an editor and reviewers for this manuscript.

As with all papers reviewed by the journal, your manuscript was reviewed by members of the editorial board and by several independent reviewers. In light of the reviews (below this email), we would like to invite the resubmission of a significantly-revised version that takes into account the reviewers' comments.

We cannot make any decision about publication until we have seen the revised manuscript and your response to the reviewers' comments. Your revised manuscript is also likely to be sent to reviewers for further evaluation.

Sincerely,

David Skurnik, M.D., Ph.D.

Section Editor

PLOS Pathogens

David Skurnik

Section Editor

PLOS Pathogens

Michael Malim

Editor-in-Chief

PLOS Pathogens

orcid.org/0000-0002-7699-2064

We are sincerely sorry for the unusual delay in finding an editor and reviewers for this manuscript.

Reviewer's Responses to Questions

**Part I - Summary**

Reviewer #1: In this manuscript the team of authors report a novel approach to develop an outer membrane vesicle vaccine from Neisseria gonorrhoeae to prevent gonorrheal infections. Critically, they inactivated the rmpM gene, which encodes an outer membrane protein that elicits the generation of undesired and detrimental blocking antibodies; note that loss of this peptidoglycan-binding protein has been reported to increase elaboration of OMVs. Secondly, they replaced the gonococcal porB1B in strain FA1090 with the meningococcal porB gene. This is noteworthy because the gonococcal PorB1B has immunosuppressive action. The platform they describe is imaginative and worthy of further study in protection-directed experiments, which I assume is on their agenda. Although there are clear strengths to the paper I do have a few concerns, which are listed below.

Reviewer #2: This study examines the possible increase in immunogenicity and potential efficacy of a gonococcal OMV vaccine that has substituted the meningococcal PorB for the gonococcal PorB. There is extensive evidence that meningococcal Porb is immuno-stimulatory via TLR2 and has been shown to act as a vaccine adjuvant. There is less evidence this property is contained within gonococcal PorB. If anything there is suggestive evidence that it could even be immuno-inhibitory They were able to show improved immunogenicity in mice with TH1 skewing and increased SBA activity.

**Part II – Major Issues: Key Experiments Required for Acceptance**

Reviewer #1: 1. The authors report outer membrane protein differences depending on whether the host FA1090 strain produces the Ng or Nm PorB. While I think the data are convincing it is important to note that OMVs also contain lipooligosaccharide (LOS). In this regard, the 2C7 epitope within LOS has been advanced by Rice et al. as a protective antigen. Thus, it would be good if the authors report whether this epitope is present in their OMVs and if they observed an antibody response to it.

2. The authors used SBA to ascertain whether the two OMVs preparations would generate different titers in mice. Was this done because they think it might be a correlate of protection? A number of points need to be be clarified: i. what % survival do they consider as bactericidal?; ii. how many mice were used to pool immune sera?; iii. did they test serum from individual mice as I suspect there could be significant variation between them?; iv. the SBA data need statistical evaluation as a two-fold difference in the titers may not be biologically important if production of bactericidal antibody is a correlate of protection; v. did the pooled sera kill other Ng strains especially those producing PorB1A or antibiotic resistant strains?

Reviewer #2: Various parameters of examining the immuno-stimulatory ability of the OMV containing meningococcal PorB were examined, including IgG measurements towards a large array of gonococcal antigens, and Bactericidal activity. A significant portion of the paper was spent in explaining the construction and examination of the OMV derived form the gonococci containing NM PorB. However though immunogenicitiy was examined, by Ab levels, possible IFNg antigen specific memory recall. and SBA, other studies should be included to help truly demonstrate potential superiority to the GC OMV. This would include demonstration of altered Ab binding to the native gonococcus. Also though there are many issues with the mouse model of infection, these studies also should be considered especially since Ann Jerse is a coauthor. THe splenocyte specific restimulation studies could have bene more sophisticated by purifying T cells and then use antigen presenting cells fed intact gonococci which then use these to stimulate the supposedly memory T cells induced by the vaccines. This would be instead of restimulation with the vaccine itself. This would give more definitive data and would be more relevant and related to potential gonococcal activity.

**Part III – Minor Issues: Editorial and Data Presentation Modifications**

Reviewer #1: The authors should indicate limitations of their work especially lack of assays that evaluated opsonophagocytic killing mediated by the pooled sera from the immunized mice.

Reviewer #2: The heat map of measurements of gonococcal IgG induced by either vaccine is hard to interpret and really doesn't display differences as compared to other possible studies like a volcano plot etc. Regarding Th1 vs Th2 responses, usually a ratio of IgG1 to IgG2a is helpful in demonstration this. This is because instead of skewing, just measuring subtype levels, increases of each one may just be due to increased overall immunity instead of true skewing. The memory T cell IFNg studies would also help in this regard,.

PLOS authors have the option to publish the peer review history of their article (what does this mean?). If published, this will include your full peer review and any attached files.

Reviewer #1: No

Reviewer #2: **Yes: **Lee M. Wetzler
---

## [Decision Letter · Decision Letter 1]

8 Oct 2024

Dear Professor Tang,

Thank you very much for submitting your manuscript "Tackling immunosuppression by Neisseria gonorrhoeae to facilitate vaccine design" for consideration at PLOS Pathogens. As with all papers reviewed by the journal, your manuscript was reviewed by members of the editorial board and by several independent reviewers. The reviewers appreciated the attention to an important topic. Based on the reviews, we are likely to accept this manuscript for publication, providing that you modify the manuscript according to the review recommendations.

Sincerely,

David Skurnik, M.D., Ph.D.

Section Editor

PLOS Pathogens

David Skurnik

Section Editor

PLOS Pathogens

Michael Malim

Editor-in-Chief

PLOS Pathogens

orcid.org/0000-0002-7699-2064

Reviewer Comments (if any, and for reference):

Reviewer's Responses to Questions

**Part I - Summary**

Reviewer #1: In this revised manuscript the authors addressed my two previous concerns dealing with their work on developing a novel OMV vaccine platform for future immunization that would protect at-risk individuals from acquiring gonorrheal infection. This work has significance for advancing public health. In general, the scholarship of the work is high and the experimental protocols that were employed are rigorous.

**Part II – Major Issues: Key Experiments Required for Acceptance**

Reviewer #1: The authors addressed my concerns regarding the 2C7 LOS epitope in the OMV preparations as well as the SBA results. In sum, the paper is much improved. However, the new information in the text regarding the 2C7 LOS epitope is a bit confusing and does not align with the information in the legend. My reading of the section indicates that the blot probed with the 2C7 monoclonal antibody is actually shown in Figure 4A while the sera from OMV-immunized ice is shown in the two panels in 4B. Interestingly, both of the OMV sera react with LOS from both WT and the lgtE::kan transformant (2C7-negative) suggesting that an antibody response to an antigen within the deep LOS core is possible. In this regard, I request that the authors re-evaluate their writing and make the necessary changes to improve clarity. Additionally, in future experiments it would be beneficial to use an lgtG deletion mutant to discriminate antibody responses the 2C7 epitope vs the inner region of the LOS alpha-chain given the possibility that this LOS epitope could stimulate production of bactericidal antiody.

**Part III – Minor Issues: Editorial and Data Presentation Modifications**

Reviewer #1: None

PLOS authors have the option to publish the peer review history of their article (what does this mean?). If published, this will include your full peer review and any attached files.

Reviewer #1: No

Figure Files:

Data Requirements:

Reproducibility:

References:

---

## [Editor Report · Decision Letter 2]

22 Oct 2024

Dear Professor Tang,

We are pleased to inform you that your manuscript 'Tackling immunosuppression by Neisseria gonorrhoeae to facilitate vaccine design' has been provisionally accepted for publication in PLOS Pathogens.

Best regards,

David Skurnik, M.D., Ph.D.

Section Editor

PLOS Pathogens

David Skurnik

Section Editor

PLOS Pathogens

Michael Malim

Editor-in-Chief

PLOS Pathogens

orcid.org/0000-0002-7699-2064
---

## [Editor Report · Acceptance letter]

4 Nov 2024

Dear Professor Tang,

We are delighted to inform you that your manuscript, "Tackling immunosuppression by Neisseria gonorrhoeae to facilitate vaccine design," has been formally accepted for publication in PLOS Pathogens.

Best regards,

Michael Malim

Editor-in-Chief

PLOS Pathogens

orcid.org/0000-0002-7699-2064